# Lateral habenula glutamatergic neurons projecting to the dorsal raphe nucleus promote aggressive arousal in mice

Aki Takahashi [1,2,3✉], Romain Durand-de Cuttoli [3], Meghan E. Flanigan[3,14], Emi Hasegawa[4,5], Tomomi Tsunematsu [6,7,8], Hossein Aleyasin [3], Yoan Cherasse [5], Ken Miya[9,10], Takuya Okada[9], Kazuko Keino-Masu[9], Koshiro Mitsui[1,10], Long Li [3], Vishwendra Patel[11], Robert D. Blitzer [11], Michael Lazarus [5], Kenji F. Tanaka [12], Akihiro Yamanaka [13], Takeshi Sakurai [4,5], Sonoko Ogawa[2] & Scott J. Russo [3]

The dorsal raphe nucleus (DRN) is known to control aggressive behavior in mice. Here, we found that glutamatergic projections from the lateral habenula (LHb) to the DRN were activated in male mice that experienced pre-exposure to a rival male mouse ("social instigation") resulting in heightened intermale aggression. Both chemogenetic and optogenetic suppression of the LHb-DRN projection blocked heightened aggression after social instigation in male mice. In contrast, inhibition of this pathway did not affect basal levels of aggressive behavior, suggesting that the activity of the LHb-DRN projection is not necessary for the expression of species-typical aggressive behavior, but required for the increase of aggressive behavior resulting from social instigation. Anatomical analysis showed that LHb neurons synapse on non-serotonergic DRN neurons that project to the ventral tegmental area (VTA), and optogenetic activation of the DRN-VTA projection increased aggressive behaviors. Our results demonstrate that the LHb glutamatergic inputs to the DRN promote aggressive arousal induced by social instigation, which contributes to aggressive behavior by activating VTA-projecting non-serotonergic DRN neurons as one of its potential targets.

[1] Laboratory of Behavioral Neurobiology, Faculty of Human Sciences, University of Tsukuba, Tsukuba, Ibaraki 305-8577, Japan. [2] Laboratory of Behavioral Neuroendocrinology, Faculty of Human Sciences, University of Tsukuba, Tsukuba, Ibaraki 305-8577, Japan. [3] Nash Family Department of Neuroscience and Brain & Body Research Center, Icahn School of Medicine at Mount Sinai, New York, NY 10029, USA. [4] Department of Molecular Behavioral Physiology, Faculty of Medicine, University of Tsukuba, Tsukuba, Ibaraki 305-8575, Japan. [5] International Institute for Integrative Sleep Medicine (WPI-IIIS), University of Tsukuba, Tsukuba, Ibaraki 305-8575, Japan. [6] Super-network Brain Physiology, Graduate School of Life Sciences, Tohoku University, Sendai, Miyagi 980-8577, Japan. [7] Advanced Interdisciplinary Research Division, Frontier Research Institute for Interdisciplinary Sciences, Tohoku University, Sendai, Miyagi 980-8578, Japan. [8] Precursory Research for Embryonic Science and Technology, Japan Science and Technology Agency, Kawaguchi, Saitama 332-0012, Japan. [9] Department of Molecular Neurobiology, Faculty of Medicine, University of Tsukuba, Tsukuba, Ibaraki 305-8575, Japan. [10] Graduate School of Comprehensive Human Sciences, University of Tsukuba, Tsukuba, Ibaraki 305-8575, Japan. [11] Department of Pharmacological Sciences and Department of Psychiatry, Icahn School of Medicine at Mount Sinai, New York, NY 10029, USA. [12] Department of Neuropsychiatry, Keio University School of Medicine, Shinjuku, Tokyo 160-8582, Japan. [13] Department of Neuroscience II, Research Institute of Environmental Medicine, Nagoya University, Nagoya, Aichi 464-8601, Japan. [14] Present address: Bowles Center for Alcohol Studies, University of North Carolina School of Medicine, Chapel Hill 27599 NC, USA. ✉email: aktakaha@human.tsukuba.ac.jp

Aggressive behavior is widely conserved in many animal species to compete for territory, food, and mates, which enhances an individual's survival and reproductive success. However, when aggressive behavior is escalated, it increases the risk of getting severely injured during the fight[1] and thus may become maladaptive. Social instigation has been shown to escalate aggressive behavior in many animal species from fish to rodents[2–4]. In these models, a short sensory encounter with a potential conspecific rival increases aggressive behavior in subsequent agonistic encounters, possibly by increasing "aggressive arousal" or "attack readiness" induced by the social instigation (also known as aggression priming)[5–8]. Aggressive arousal provoked by social instigation has been shown to intensify aggressive behavior but not other behaviors such as locomotor activity, food intake, or sexual behavior[5]. Understanding the neural mechanisms of instigation-heightened aggression will provide important insight into the biological basis of escalated aggression.

The brain serotonergic system has long been implicated in escalated aggressive behavior across many animal species[9–12]. The dorsal raphe nucleus (DRN) contains the largest population of serotonin (5-HT) neurons and is known to control aggressive behaviors[13]. Previously, we showed that the DRN plays an important role in instigation-heightened aggression in male mice[14]. In vivo microdialysis revealed an increase of glutamate release in the DRN during social instigation, and microinjection of L-glutamate into the DRN escalated aggressive behavior of male mice[14]. In addition, a separate study showed that glutamate inputs to the DRN regulates the expression of maternal aggression[15]. The DRN receives glutamatergic projections from many brain areas, including the prefrontal cortex, hypothalamic areas, the extended amygdala, and the lateral habenula (LHb)[16–19].

In this study, we show that glutamatergic projections from the LHb to the DRN are specifically involved in the escalation of intermale aggression induced by social instigation, but not the expression of species-typical aggressive behavior. We found that these LHb neurons synapse on DRN neurons that project to the ventral tegmental area (VTA), and optogenetic manipulation of DRN-VTA neurons also regulated escalated aggression. Thus, our study identifies a LHb-DRN projection—along with its potential downstream VTA-projecting non-serotonergic DRN neurons—that regulates aggressive arousal induced by social instigation.

## Results

### Social instigation-heightened aggression and LHb-DRN projection.
In this study, we examined intermale aggressive behaviors of ICR male mice in both the resident-intruder (RI) test and social instigation (Inst) test (Fig. 1a). In the Inst test, a resident male mouse was exposed to an instigator male placed in a protective cage for 5 min, which allowed the resident to see, smell, and partially touch the instigator within his homecage but he could not physically attack it. In the following aggressive encounter, resident males in the Inst group showed a longer duration of aggressive behaviors than the RI group (Fig. 1b), confirming a previous report of instigation-heightened aggression[4]. This effect was specific to aggressive components of social behaviors including attack bites, tail rattles, and sideways threats as well as reduced attack latency (Supplementary Fig. 1). Escalation of aggressive behaviors by social instigation was observed in the first 2 min of the aggressive encounter and then returned to the level of the RI group (Fig. 1c, d). This temporal pattern of the aggression-heightening effect of social instigation was consistent with seminal work from cichlid fish[2].

Next, given our previous work showing that the DRN was involved in instigation-heightened aggression[14], we wanted to define the cell types within the DRN as well as possible projections into the DRN that might be involved. We injected RetroBeads into the DRN and found RetroBead-labeled neurons in the LHb and lateral hypothalamus (LH) (Fig. 1e, Supplementary Fig. 2). We first analyzed the activation of LHb neurons by an aggressive encounter using c-Fos immunohistochemistry (Fig. 1f–h). Both right and left hemispheres of the LHb from 6 consecutive slices with a 90 μm interval were analyzed from each animal in control (Cont, $n = 8$ animals), RI ($n = 8$ animals), and Inst ($n = 10$ animals) groups. The average number of c-Fos or Retrobead-labeled cells and their % colocalization in the LHb per slice were calculated in each animal. An aggressive encounter caused an increase in c-Fos expression in the LHb in both the RI and Inst groups compared to the Cont group ($43.9 \pm 8.3$ cells for Cont, $93.7 \pm 8.8$ cells for RI, $100.2 \pm 4.9$ cells for Inst; Fig. 1i). While the number of RetroBead-labeled cells were not different among all three groups ($48.4 \pm 15.8$ cells for Cont, $52.6 \pm 12.5$ cells for RI, $56.1 \pm 14.6$ cells for Inst; Fig. 1j), c-Fos colocalization within RetroBead-labeled DRN-projecting LHb (LHb-DRN) neurons was higher in the Inst group compared to the Cont group ($13.2 \pm 2.6\%$ for Cont, $21.6 \pm 3.3\%$ for RI, $33.5 \pm 6.2\%$ for Inst; Fig. 1k), whereas the RI group did not show increased activation of LHb-DRN neurons compared to the Cont group. There was no side difference between right and left LHb in the total number of c-Fos and Retrobead-labeled cells (Supplementary Table 1). LHb-DRN projection neurons have been identified as glutamatergic[17–19], and we confirmed that 99.2% of RetroBead-labeled cells were colocalized with $Vglut2$ (245 cells among 247 cells analyzed; Fig. 1l–o).

To confirm whether the increased c-Fos expression in LHb-DRN neurons reflected differences in neural activity between the RI and Inst groups, we used ex vivo whole-cell patch-clamp electrophysiology recording (Fig. 2a) to measure spontaneous firing rate and resting membrane potential. We first injected a retrograde AAV-EGFP (AAVretro-hSyn-EGFP (referred to below as AAVretro-EGFP)) into the DRN. The RI test ($n = 5$) or Inst test ($n = 5$) was conducted 4–6 weeks after the injection. Thirty minutes after the aggressive encounter, brains were removed, sliced, and ex vivo recordings were obtained from the EGFP + LHb neurons. We found that the number of LHb-DRN neurons exhibiting spontaneous activity was higher in the Inst group compared to the RI group (8 cells out of 25 cells (32%) for Inst, 2 cells out of 24 cells (8.3%) for RI; Fig. 2b), and the average spontaneous firing rate was significantly higher in the Inst group compared to the RI group (Fig. 2c). In addition, there was a significant difference in the resting membrane potential (RMP) with the Inst group depolarized relative to the RI group (average −62.1 mV for Inst and −67.1 mV for RI; Fig. 2d, e). These results indicate that social instigation-heightened aggression is associated with higher activity of LHb neurons that project to the DRN.

We also analyzed c-Fos expression in RetroBead-labeled cells in the LH, and DRN-projecting LH cells and found they were activated by an aggressive encounter in both RI and Inst groups relative to Cont groups, but they did not differ from one another (Supplementary Fig. 2). From these data, we conclude that the LHb-DRN projection may be involved in aggressive arousal induced by social instigation in male mice.

### Inhibition of LHb-DRN projection prevents escalation of aggression.
To examine if the LHb-DRN projection is involved in the escalation of aggressive behaviors due to social instigation, we performed chemogenetic and optogenetic inhibition of this projection during social instigation. For chemogenetic inhibition of the LHb-DRN projection, we injected retrograde AAV-Cre (AAVretro-EF1α-mCherry-IRES-Cre (referred to below as AAVretro-Cre)) into the DRN and a Cre-dependent AAV2-CAG-Flex-rev-hM4D-2A-

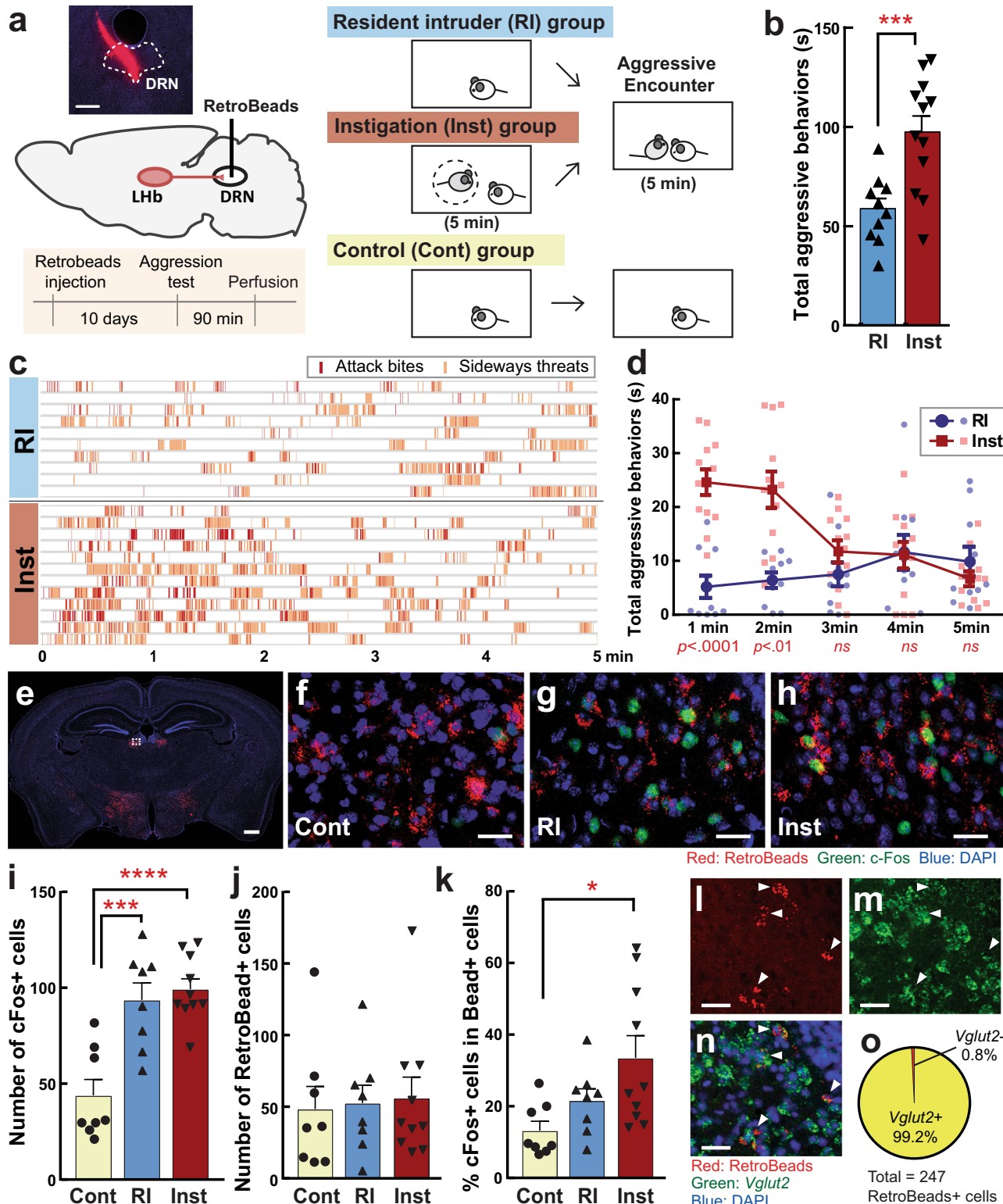

Red: RetroBeads   Green: c-Fos   Blue: DAPI

Red: RetroBeads
Green: *Vglut2*
Blue: DAPI

EGFP (referred below as hM4D), or Cre-dependent AAV2-EF1α-DIO-ChR2(H134R)-EYFP (referred below as EYFP) as a control, into bilateral LHb (Fig. 3a). Specific localization of GFP+ neurons was observed in the LHb (Fig. 3b, c). Three weeks after AAV injection, we tested animals in the RI test and Inst test after i.p. injection of either saline (SAL) or 1 mg/kg clozapine-N-oxide (CNO; Fig. 3a). As expected, we found that social instigation increased the duration of total aggressive behaviors regardless of

SAL or CNO injection compared to the RI test in EYFP control animals (Fig. 3d, black line). However, CNO injection into the hM4D males in the Inst test blocked the pro-aggressive effect of social instigation (Fig. 3d, green line). Average data showed that the control EYFP group exhibited an increase in the duration of total aggressive behaviors in the Inst tests with both SAL and CNO injection compared to the RI test, whereas the hM4D group showed instigation-heightened aggression only with SAL but not after CNO

**Fig. 1 Social instigation-heightened aggression and c-Fos in LHb-DRN projection neurons. a** Schematics of this experiment. Test animals were injected with RetroBeads into the DRN (outlined in white lines; scale bar 400 μm). The standard resident-intruder (RI) group was tested for 5 min during RI test. The social instigation (Inst) group had a 5 min exposure to a caged-instigator male prior to the 5 min RI test. Control (Cont) animals were kept undisturbed. **b** Inst group showed longer duration of aggressive behaviors compared to RI group (two-way repeated measures ANOVA, RI $n = 10$, Inst $n = 12$ biologically independent animals, main effect of Group: F(1,20) = 16.93, $p = 0.0005$). **c** Temporal pattern of occurrence of attack bites (red) and sideways threat (orange) in individual animals of RI and Inst groups. **d** Aggressive behaviors were increased by social instigation in the first 2 min of aggressive encounter compared to RI group (two-way repeated-measures ANOVA with the Geisser-Greenhouse correction, RI group $n = 10$ and Inst group $n = 12$ biologically independent animals, Group × time interaction, F(4,80) = 9.615, $p < 0.0001$, post hoc $t$ test with Bonferroni's correction (two-sided)). **e** RetroBead+ cells (red) were observed in the LHb and LH (scale bar 500 μm). Higher magnification of white-dotted square is indicated in **f–h** (scale bar 20 μm). c-Fos expression (green) and RetroBead+ cells (red) in the LHb of Cont (**f**), RI (**g**), and Inst (**h**) groups. Blue: DAPI (scale bar 20 μm). Average number of c-Fos expressing cells (**i**) and RetroBead+ cells (**j**) in the LHb per slice. Both RI and Inst groups showed higher number of c-Fos expressing cells in the LHb compared to Cont (**i**, one-way ANOVA, Cont $n = 8$, RI $n = 8$, Inst $n = 10$ biologically independent animals, F(2,24) = 18.00, $p < 0.0001$, post hoc $t$ test with Tukey's multiple comparisons test (two-sided)). **j** Number of RetroBead+ cells in the LHb were not different among groups (**j**, one-way ANOVA, Cont $n = 8$, RI $n = 8$, Inst $n = 10$ biologically independent animals, n.s.). **k** Percent of c-Fos expressing RetroBead+ cells were significantly higher in Inst group compared to Cont (Kruskal–Wallis test (two-sided), Cont $n = 8$, RI $n = 8$, Inst $n = 10$ biologically independent animals, Kruskal–Wallis statistic = 7.638, $p = 0.0219$, post hoc test with Dunn's multiple comparisons test (two-sided)). Representative picture of RetroBead+ cells (**l**), *Vglut2* positive cells (**m**), and their overlay with DAPI (**n**) in the LHb (scale bar 20 μm) stained by in situ hybridization. Arrowhead indicates RetroBead and *Vglut2* colocalized cells. **o** Percent of RetroBead+ cells ($n = 247$ biologically independent cells) that colocalized with *Vglut2*. *$p < 0.05$, ***$p < 0.001$, ****$p < 0.0001$. Error bars indicate standard error of the mean (SEM). Source data are provided as a Source Data file.

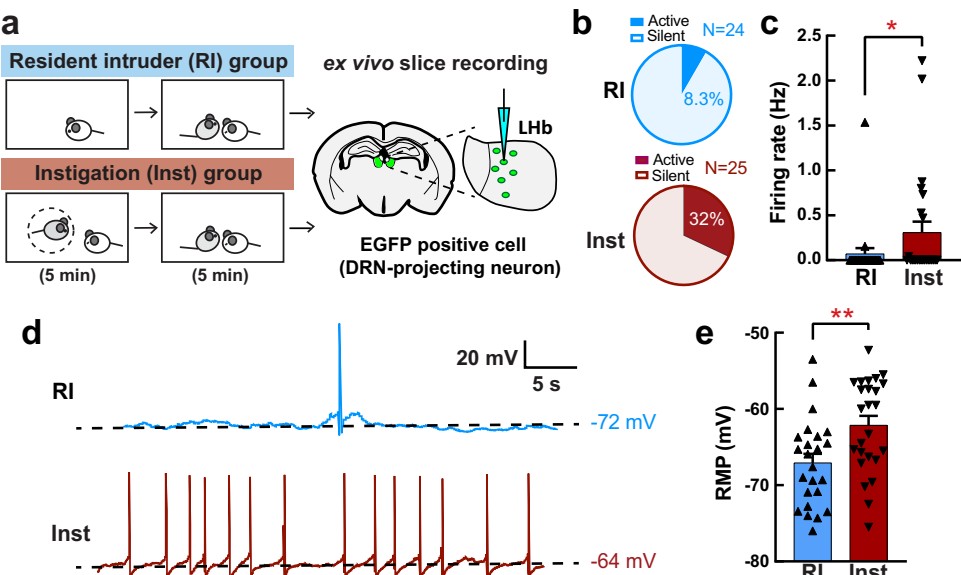

**Fig. 2 Social instigation-heightened aggression is associated with increased spontaneous activity in LHb-DRN projection neurons. a** Schematics of the experiment. Animals were injected a retrograde AAVretro-hSyn-EGFP into the DRN at least 4 weeks before the test. On the test day, RI group ($n = 5$ biologically independent animals) was tested for 5 min RI test, and Inst group ($n = 5$ biologically independent animals) had 5 min exposure to a caged-instigator male prior to 5 min aggression test. Recording was conducted from the EGFP+ cells in the LHb (3–6 neurons per mouse). **b** Percentage of the recorded neurons that showed spontaneous firing (Active) was higher in the Inst group (bottom: 8 Active cells out of 25 recorded cells) than the RI group (top: 2 Active cells out of 24 recorded cells) (Chi-square test, $X^2$ (1,49) = 4.222, $p = 0.0399$). **c** Average firing rate was higher in the Inst group compared to RI group (Mann-Whitney test (two-sided), RI $n = 25$, Inst $n = 24$ biologically independent cells, U = 228, $p = 0.0500$). **d** Representative traces of spontaneous firing pattern of the EGFP + LHb neurons of RI animal (top, blue line) and Inst animal (bottom, red line). Black dotted lines indicate the resting membrane potential (RMP) of that cell. **e** Average RMP (mV) was higher in the Inst group compared to the RI group (Unpaired $t$ test with Welch's correction (two-sided), RI $n = 25$, Inst $n = 24$ biologically independent cells, t(46.98) = 2.906, $p = 0.0056$). *$p < 0.05$, **$p < 0.01$. Error bars indicate standard error of the mean (SEM). Source data are provided as a Source Data file.

injection (Fig. 3e). Detailed behavioral analysis showed a similar effect on the frequency of attack bites (Fig. 3f). Both Inst groups (SAL + Inst and CNO + Inst) showed a higher frequency of attack bites compared to the SAL + RI in the EYFP control group, whereas CNO injection reduced attack bite frequency after social instigation (CNO + Inst) and there was a significant difference between SAL + Inst and CNO + Inst in the hM4D group (Fig. 3f). No group difference was observed in other indices of aggressive behaviors or non-aggressive behaviors (Fig. 3g–i, Supplementary Fig. 3).

After the last Inst test, we conducted two additional RI tests with SAL and CNO injections to examine the effect of LHb-DRN inhibition on aggression. There was no effect of CNO injection on either aggressive or non-aggressive behaviors in the EYFP control or LHb-DRN hM4D groups in the RI test (Fig. 3j–n, Supplementary Fig. 3). Therefore, the aggression-suppressive effect of LHb-DRN inhibition is restricted to instigation-heightened aggression.

We also inhibited LHb-DRN neurons optogenetically by expressing eNpHR3.0 (AAV2-hSyn-eNpHR3.0-EYFP) in the

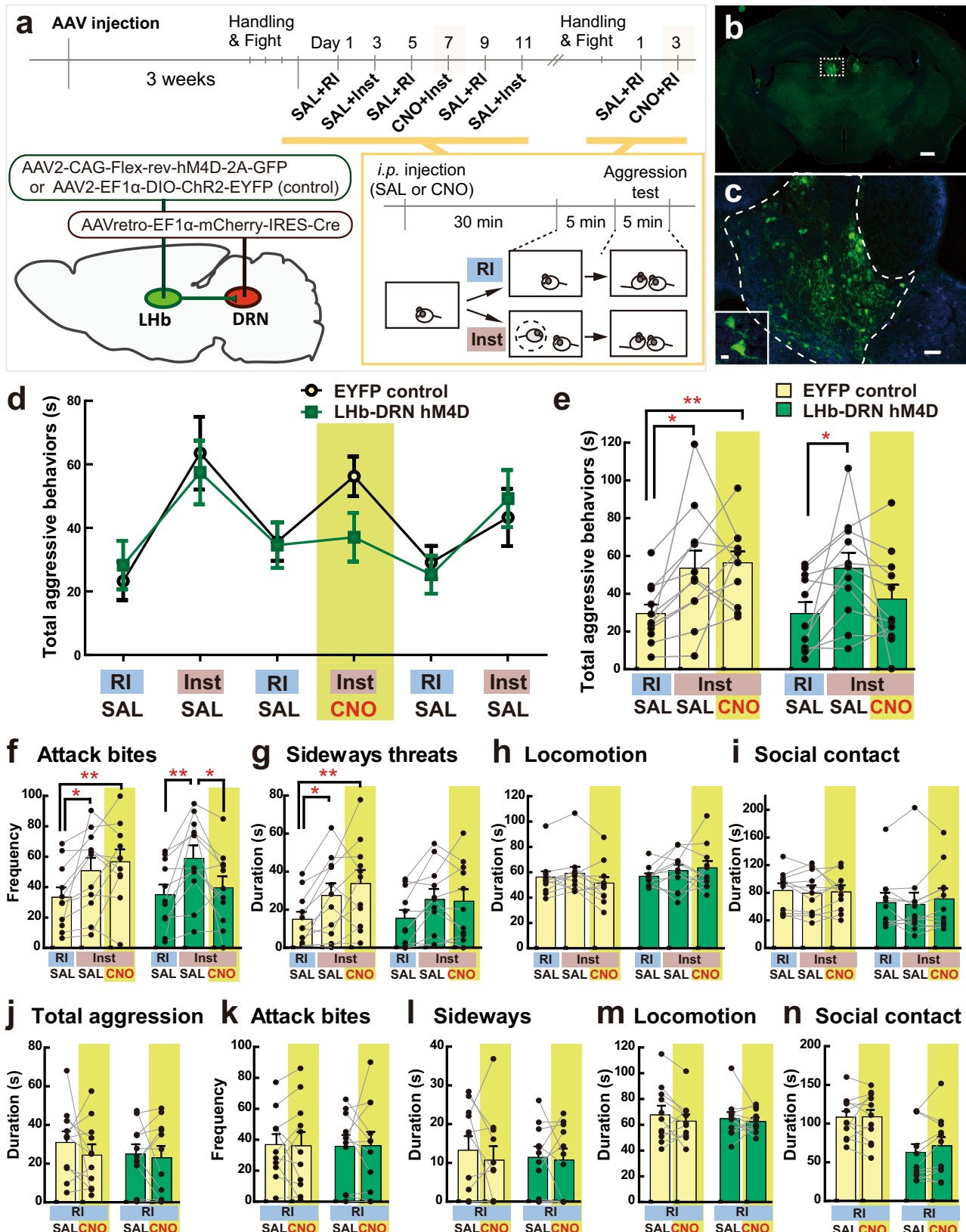

LHb and then applying yellow light (593 nm, 3 mW) to the terminals in the DRN (Fig. 4a–c). Intra-DRN light was delivered from the beginning of social instigation until the end of the aggressive encounter. Again, we observed that inhibition of LHb-DRN projection blocked the escalation of aggressive behaviors by social instigation. As expected, social instigation without light

illumination (Inst+OFF) increased aggressive behaviors compared to the RI test. Light illumination during social instigation (Inst+ON) suppressed aggressive behaviors to the same level as that of the standard RI test for the total duration of aggressive behaviors (Fig. 4d, e), frequency of attack bites (Fig. 4f), duration of sideways threats (Fig. 4g), tail rattles and pursuits

**Fig. 3 Chemogenetic inhibition of LHb-DRN projection neurons and instigation-heightened aggression. a** Schematics and timeline of this experiment. Retrograde AAVretro-EF1α-mCherry-IRES-Cre was injected into the DRN, and Cre-dependent AAV2-CAG-Flex-rev-hM4D-2A-GFP (or control AAV2-EF1α-DIO-ChR2-EYFP) was injected into bilateral LHb. Aggression tests were conducted every other day, and the RI test and Inst test were conducted alternately. CNO was injected only on Day 7, and saline (SAL) was injected all other days. Then, animals were tested in the RI test with SAL and CNO injections. **b, c** Expression of hM4D-EYFP (green) was observed in bilateral LHb (**b** scale bar 500 μm). Higher magnification of white-dotted square is indicated in **c** (scale bar 50 μm). Inserted picture shows an enlarged picture of the LHb (outlined in white lines, scale bar 10 μm). **d** Duration of total aggressive behaviors in each session. Black line (yellow circles) indicates average of EYFP control ($n = 11$ biologically independent animals), and green line (green squares) indicates average of hM4D group ($n = 11$ biologically independent animals, Mean ± SEM). **e** Summarized data for the duration of total aggressive behavior in each condition. Social instigation increased total aggressive behavior regardless of SAL or CNO injection compared to RI in EYFP controls, but the CNO + Inst group did not show significant differences from the SAL + RI session of the LHb-DRN hM4D group (two-way repeated measures ANOVA, Cont $n = 11$, hM4D $n = 11$ biologically independent animals, main effect of test-type: $F_{(2,40)} = 9.599$, $p = 0.0004$, post hoc $t$ test with Tukey's multiple comparisons tests (two-sided)). **f–i**, Detailed behavioral analysis showed that CNO blocked pro-aggressive effects of social instigation, specifically in the hM4D group, on the frequency of attack bites (**f** two-way repeated measures ANOVA, Cont $n = 11$, hM4D $n = 11$ biologically independent animals, Group × Test-type interaction: $F_{(2,40)} = 3.877$, $p = 0.0289$, post hoc $t$ test with Tukey's multiple comparisons tests (two-sided)). No Group × Test-type interaction was observed in sideways threats (**g**), locomotion (**h**) or social contacts (**i**) (two-way repeated-measures ANOVA, Cont $n = 11$, hM4D $n = 11$ biologically independent animals, Group × Test-type interaction: all n.s., main effect of test-type: sideways $F_{(2,40)} = 12.04$, $p < 0.0001$ (**g**), locomotion and social contact n.s. (**h, i**), post hoc $t$ test with Tukey's multiple comparisons tests (two-sided)). No effect of CNO was observed in the RI test on the duration of total aggressive behavior (**j**), frequency of attack bites (**k**), duration of sideways threats (**l**), locomotion (**m**), and social contacts (**n**) (two-way repeated-measures ANOVA, Cont $n = 11$, hM4D $n = 11$ biologically independent animals, all n.s.). Each bar represents mean value ± SEM, and gray line indicates each individual's data. $*p < 0.05$, $**p < 0.01$. Source data are provided as a Source Data file.

(Supplementary Fig. 4). Indeed, significant differences between Inst+OFF and Inst+ON were observed in the attack bites and sideways threats (Fig. 4f, g). In contrast, no effect of social instigation nor light delivery was observed in any non-aggressive behaviors (Fig. 4h, i). We also did not observe any effects of light delivery on aggressive and non-aggressive behaviors in the RI test without social instigation (Fig. 4j–n, Supplementary Fig. 4).

These results indicate that suppression of the LHb-DRN projection blocks the pro-aggressive effect of social instigation, indicating that the activation of the LHb-DRN projection is necessary for instigation-heightened aggression. Notably, inhibition of the LHb-DRN projection does not affect aggressive behavior in the standard RI test. Thus, this pathway is specifically required for the escalation of aggression by social instigation, but not the expression of species-typical levels of aggressive behaviors.

**Activation of the LHb-DRN projection escalates aggressive behaviors.** To examine if the activation of the LHb-DRN projection is sufficient to escalate aggressive behaviors, we optogenetically activated the LHb-DRN projection by expressing ChR2 in the LHb and delivering light to the terminals in the DRN (Fig. 5a). A ChR2-expressing AAV (AAV2-hSyn-hChR2(H134R)-EYFP (referred below as ChR2)) was injected into bilateral LHb (Fig. 5b), and EYFP+ projections were visualized in the DRN where the optic fiber was inserted (Fig. 5c). To examine the effect of LHb-DRN stimulation, we examined c-Fos expression in the DRN after 1 min of light stimulation (473 nm, 20 Hz, 10-ms pulse, 3 mW: Light ON $n = 8$) or no light stimulation (Light OFF $n = 6$) without any aggressive encounter. We confirmed that light stimulation of LHb-DRN terminals caused an increase in c-Fos expression (Fig. 5d) in both serotonergic neurons (Light OFF: 20.8 ± 4.0 cells, Light ON: 52.1 ± 4.7 cells per slice; Fig. 5e) and non-serotonergic neurons (Light OFF: 38.3 ± 11.0 cells, Light ON: 120.0 ± 14.2 cells per slice; Fig. 5f) in the DRN. This was consistent with reports that the LHb sends glutamatergic projections to both serotonergic neurons and GABAergic neurons in the DRN[18,19]. Five weeks after AAV injection, we examined aggressive behavior under 3 types of light stimulation schemes (Fig. 5g). To mimic the social instigation procedure, where the resident male was exposed to an instigator prior to the aggressive encounter, we stimulated the LHb-DRN projection 1 min before introduction of an intruder. This light stimulation was terminated right after introduction of the

intruder (ON/OFF), or continued until the end of the aggressive encounter (ON/ON). In addition, we conducted a simple ON session in which light stimulation was applied only during the RI test. All light stimulation sessions were flanked by an OFF session in which the RI test was conducted without any light stimulation (Fig. 5h). We found that both ON and ON/ON stimulation schemes increased aggressive behaviors compared to OFF sessions, and the ON/ON session showed the strongest pro-aggressive effect (Fig. 5i, j). Detailed behavioral analysis showed that the ON/ON scheme promoted higher aggressive behaviors compared to other stimulation schemes, including attack bites (Fig. 5k), sideways threats (Fig. 5l), and tail rattles (Supplementary Fig. 5). No effect of stimulation was observed for most of the non-aggressive behaviors except time in social contact (Fig. 5m, n). As a testament to the robustness and reproducibility of this effect, we replicated this finding in a separate group of mice within a different animal facility using just unilateral LHb-DRN ChR2 stimulation (Supplementary Fig. 6).

Although the ON/OFF session was expected to mimic the social instigation procedure, we did not observe a significant increase in aggressive behavior by this stimulation scheme. We hypothesized that the LHb-DRN activation might need to be paired with a social stimulus to increase aggressive behavior. To examine this possibility, we conducted a subthreshold social instigation test for 1-min (S-Inst; Fig. 5o) that was not sufficient to produce any aggression-heightening effect (Fig. 5p, S-Inst+OFF), unless it is combined with LHb-DRN light stimulation. With this subthreshold protocol, we found that the combination of a 1-min social instigation with optogenetic LHb-DRN stimulation (S-Inst + ON/OFF) significantly increased aggressive behaviors compared to RI or S-Inst+OFF (Fig. 5p, q, Supplementary Fig. 7). To determine whether the LHb-DRN circuit regulates other forms of aggressive behavior, we performed LHb-DRN stimulation during female interaction and found no effect (Fig. 5r). This stimulation also did not induce sexual behavior such as mounting or intromissions toward the female nor did it affect spontaneous locomotor activity in the homecage (Supplementary Fig. 7). These results indicate that activation of the LHb-DRN projection specifically escalates intermale aggressive behavior when it is combined with the male opponent. In summary, our current findings show that increased activation of the LHb-DRN glutamate projection promotes the escalation of aggressive behaviors by social instigation.

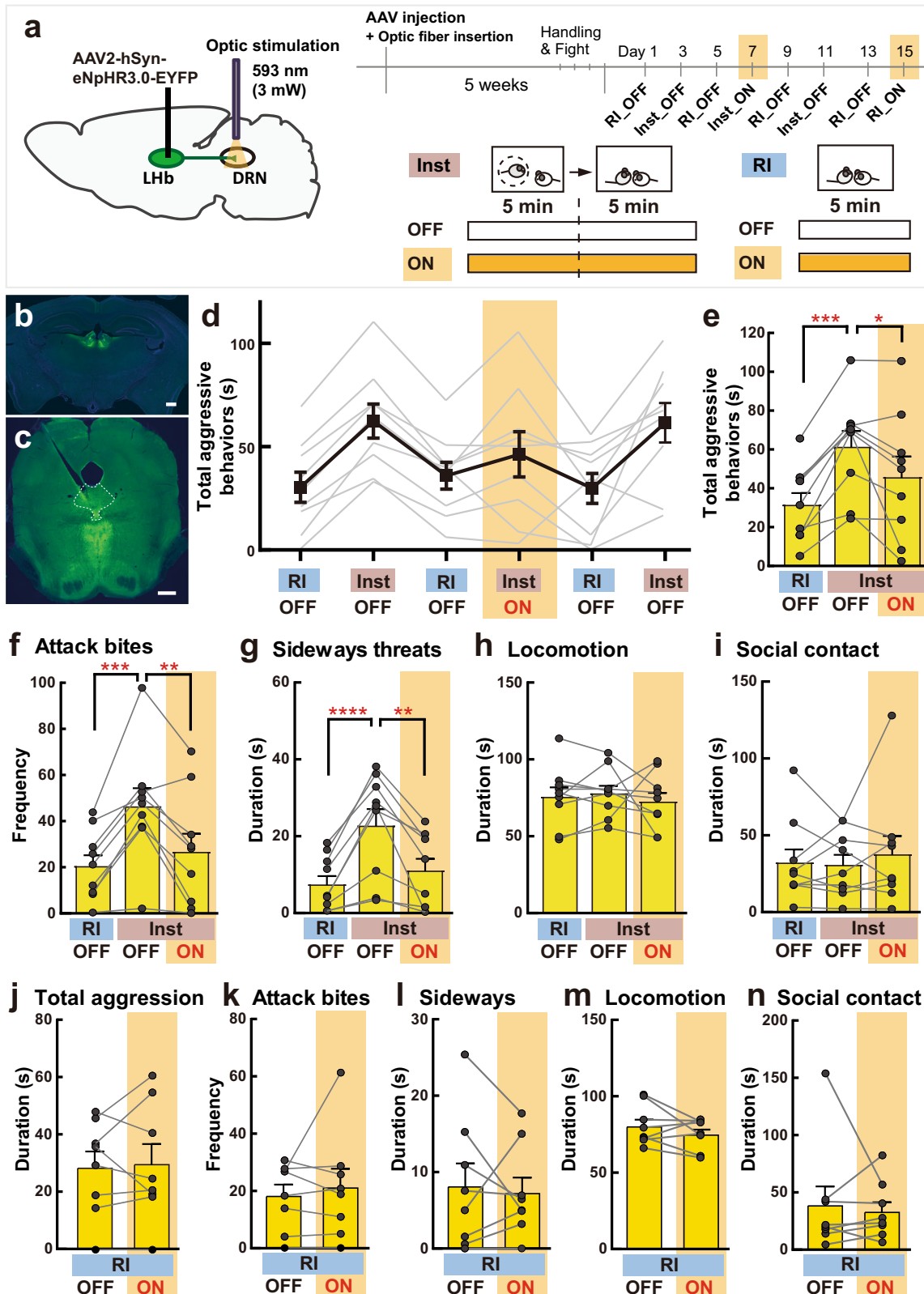

**Suppression of DRN 5-HT neuron does not affect instigation aggression**. We next examined which DRN neurons are responsible for the aggression-promoting effect of activation of the LHb-DRN glutamatergic input. The DRN contains the largest number of 5-HT neurons that project to forebrain areas[20]. Therefore, we examined the possible involvement of DRN 5-HT neurons on aggressive behaviors by using tryptophan hydroxylase

2 (Tph2)-tTA::TetO-eArchT-EYFP transgenic mice[21] to inhibit 5-HT neuron activity during aggressive encounters. Due to low aggressive behaviors in the original genetic background of this transgenic mouse, we crossed these mice with ICR/CD-1 females to produce F1 offspring that have both Tph2-tTA and TetO-eArchT-EYFP loci. Inhibition of DRN 5-HT neurons was confirmed using patch-clamp recordings in brain slices. Yellow light

**Fig. 4 Optogenetic inhibition of LHb-DRN projection neurons and instigation-heightened aggression. a** Schematics and timeline of this experiment. AAV2-hSyn-eNpHR3.0-EYFP was injected into bilateral LHb. Aggression tests were conducted every other day, and RI test and Inst test was conducted alternately. Yellow light (593 nm, 3 mW) was delivered to the DRN on days 7 and 15. **b**, **c** Expression of eNpHR3.0-EYFP (green) was observed in bilateral LHb (**b** scale bar 500 μm). Blue: DAPI. Projection terminals were observed in the DRN (outlined in white lines) and median raphe nucleus (**c** scale bar 500 μm). **d** Duration of total aggressive behaviors in each session. Gray lines show each individual's data ($n = 9$ biologically independent animals) and black line indicates average data (Mean ± SEM). **e** Summarized data for the duration of total aggressive behavior in each condition. The Inst test increased total aggressive behavior compared to the RI test in OFF sessions, but light stimulation (ON) blocked the effect of social instigation (one-way repeated measures ANOVA, $n = 9$ biologically independent animals, $F_{(2,16)} = 12.65$, $p = 0.0005$, post hoc $t$ test with Tukey's multiple comparisons tests (two-sided)). **f–i** Detailed behavioral analysis showed that optogenetic inhibition of LHb-DRN blocked the pro-aggressive effect of social instigation (one-way repeated measures ANOVA, $n = 9$ biologically independent animals, attack bites: $F_{(2,16)} = 15.47$, $p = 0.0002$ (**f**), sideways threats: $F_{(2,16)} = 18.15$, $p < 0.0001$ (**g**), post hoc $t$ test with Tukey's multiple comparisons tests (two-sided)), but had no effect on non-aggressive behaviors such as locomotion (**h**) and social contact (**i**) (one-way repeated measures ANOVA, $n = 9$ biologically independent animals, both *n.s.*). No effect of light stimulation was observed in the RI test on the duration of total aggressive behavior (**j**), frequency of attack bites (**k**), duration of sideways threats (**l**), locomotion (**m**), and social contacts (**n**) (paired $t$ test (two-sided), $n = 9$ biologically independent animals, all *n.s.*). Each bar represents mean value ± SEM, and gray line indicates each individual's data. *$p < 0.05$, **$p < 0.01$, ***$p < 0.001$, ****$p < 0.0001$. Source data are provided as a Source Data file.

illumination hyperpolarized DRN 5-HT neurons and suppressed spontaneous activity (Fig. 6a–c). Even a long-term light illumination of eArchT+ serotonergic neurons (10 min) successfully suppressed neuronal activity throughout the light illumination, and firing rate returned to baseline level after the termination of the light (Fig. 6a bottom). eArchT-EYFP expression was observed in both the DRN and median raphe nucleus, though spatial localization was achieved by placing the optic fiber specifically into the DRN with a 26° angle (Fig. 6d). We examined the effect of DRN 5-HT neuron inactivation during aggression tests (RI and Inst) with the same timeline and light delivery schemes as in Fig. 4a. Under these conditions, social instigation caused increases in aggressive behaviors in both OFF and ON sessions (Fig. 6e). Both Inst+OFF and Inst+ON sessions showed significant increases in the total duration of aggressive behaviors (Fig. 6f), the frequency of attack bites (Fig. 6g), duration of tail rattles (Fig. 6h) as well as reduced attack latency (Supplementary Fig. 8) compared to the RI test. No effect of light delivery was observed in aggressive or non-aggressive behaviors in the Inst test (Fig. 6e–j), nor in the RI test (Fig. 6k–o). These results indicate that global inhibition of the DRN 5-HT neurons does not block the aggression-heightening effect of social instigation.

**LHb neurons project mainly to non-serotonergic DRN neurons.** Because our results suggest that activity of 5-HT neurons in the DRN do not promote instigation-heightened aggression, we next examined which DRN neurons projecting to specific downstream regions could be responsible for the aggression-promoting effect of the LHb-DRN glutamatergic input. To examine direct connections between the LHb and the DRN, we injected AAV1-CMV-Cre (AAV serotype 1, referred below as AAV1-Cre), which has a property of anterograde transmission[22], unilaterally into the LHb. We also injected a Cre-dependent mCherry-expressing AAV (AAV2-hSyn-DIO-mCherry, referred below as mCherry) into the DRN (Fig. 7a). Tph2+ 5-HT neurons were visualized by using Tph2-tTA::TetO-eArchT-EYFP transgenic mice. Five weeks after viral infection, mCherry+ neurons were observed in the DRN (Fig. 7b–f). Of the 195 cells that expressed mCherry in the DRN, only 29 cells (14.9%) co-expressed eArchT-EYFP ($n = 3$ animals; Fig. 7g), indicating that the major population of DRN cells that receive input from the LHb were non-serotonergic. We then examined the projection targets of those neurons by injecting AAV1-Cre into the LHb and a Cre-dependent EYFP-expressing AAV (AAV2-EF1α-DIO-NpHR3.0-EYFP, referred below as EYFP) into the DRN (Fig. 7h). Again, we confirmed that only a small population of EYFP + cells in the DRN were Tph2-immunoreactive (5.3% (24 cells out of 477

EYFP + cells: $n = 3$ animals), Supplementary Fig. 9a–e). Also, EYFP and tyrosine hydroxylase (TH) co-expression was very low (1.7% (9 cells out of 548 EYFP+ cells: $n = 3$ animals), Supplementary Fig. 9f–j). By contrast, in situ hybridization showed that 43.8% of EYFP+ cells were colocalized with vesicular glutamate transporter 3 (*Vglut3*) mRNA (56 cells out of 128 EYFP+ cells: $n = 3$ animals, Supplementary Fig. 9k–p). The EYFP+ projections were observed in the ventral tegmental area (VTA, Fig. 7i, j), bilateral LHb (Fig. 7k), medial mammillary nucleus, caudal and lateral subdivision of interpeduncular nucleus, and sparsely in the lateral hypothalamus (Supplementary Fig. 9q–s).

**Activation of the DRN-VTA projection escalates aggressive behaviors.** Because the dopamine system has been implicated in escalated aggression as well as aggression reward[23–25], we decided to study the DRN-VTA projection as one of the potential downstream targets mediating instigation-heightened aggression. To examine whether the DRN-VTA projection is involved in escalation of aggressive behaviors of male mice, we optogenetically activated the DRN-VTA projection before and during the RI test (Fig. 8a; same stimulation scheme as in Fig. 5). Either a ChR2-expressing AAV (AAV2-hSyn-ChR2-EYFP, referred below as ChR2) or a control EYFP-expressing AAV (AAV2-hSyn-EYFP, referred below as EYFP) was injected into the DRN, and an optic fiber was inserted in the VTA (Fig. 8b, c). We confirmed that 1-min of DRN-VTA ChR2 stimulation (473 nm, 20 Hz, 10-ms pulse, 3 mW: ON $n = 6$ animals) caused a significant increase of c-Fos expression in the VTA compared to no light stimulation animals (OFF $n = 6$ animals; Fig. 8d). We then examined the effect of light stimulation on aggressive behaviors using ON, ON/OFF, and ON/ON schemes described above (Fig. 8e–h). We found that all DRN-VTA ChR2 stimulation schemes increased the frequency of attack bites compared to the OFF scheme (Fig. 8g), while only ON and ON/OFF stimulation showed significant increases in the duration of total aggressive behaviors (Fig. 8f) and tail rattles (Fig. 8h). There was no effect of light stimulation on non-aggressive behaviors except locomotion in which there were significant differences between ON and ON/ON (Fig. 8i, j, Supplementary Fig. 10). In addition, no effect of light stimulation was observed in the DRN-VTA EYFP control group (Supplementary Fig. 11). Furthermore, we confirmed the aggression-heightening effect of DRN-VTA ChR2 stimulation in a separate group of mice within a different animal facility (Supplementary Fig. 12). Thus, our results indicate that activation of VTA-DRN projection has pro-aggressive effects in male mice and escalates aggressive behavior from their species-typical RI behavior.

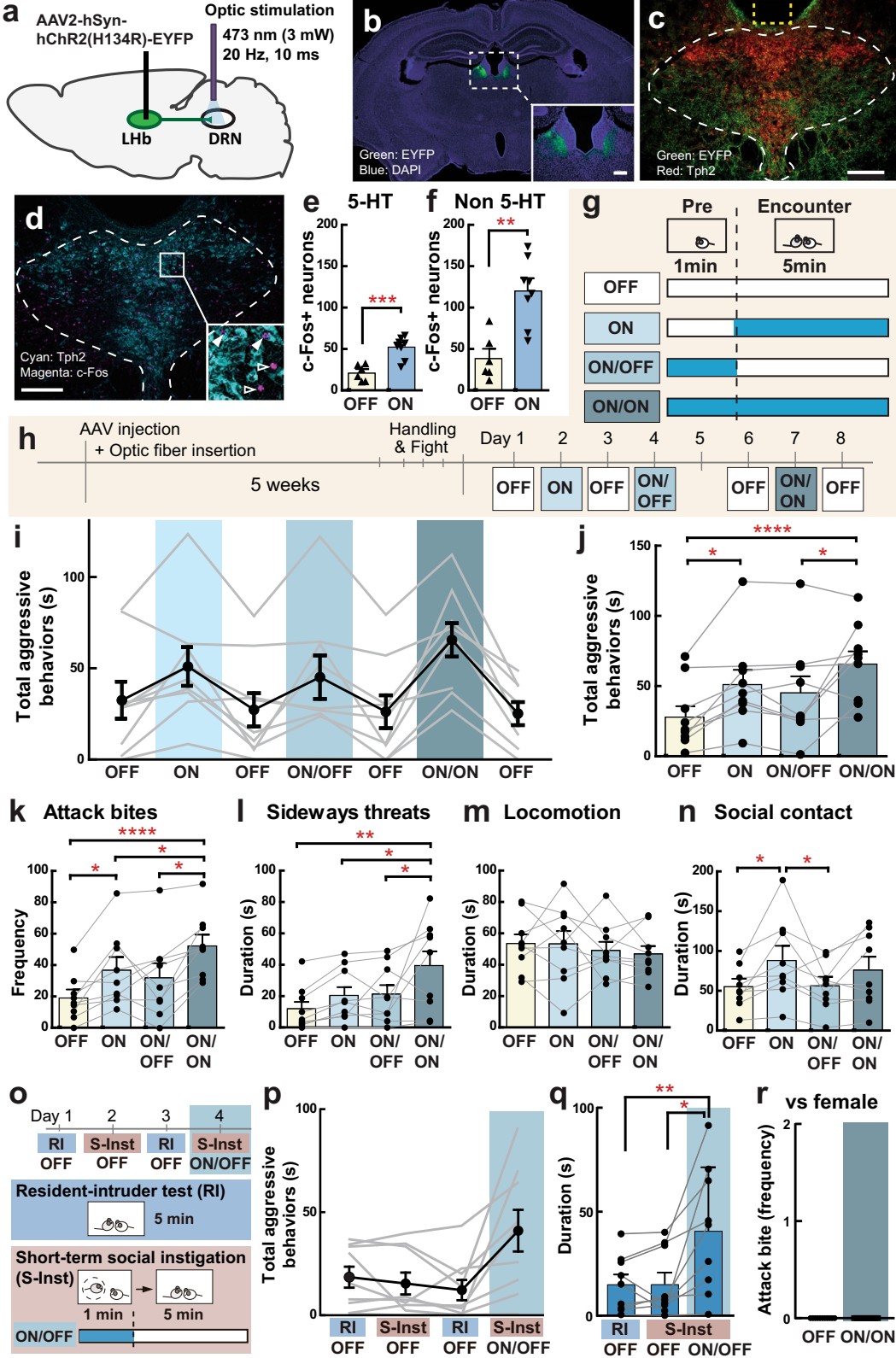

## Discussion

Previously, we showed that glutamate signaling in the DRN is associated with the escalation of intermale aggressive behavior[14], and the present study identifies at least one important source of DRN glutamate from the LHb as promoting aggressive behavior. Importantly, our results show that the LHb-DRN projection is specifically involved in the escalation of aggression due to social instigation but not the expression of species-typical aggressive behavior. Both chemogenetic and optogenetic inhibition of LHb-DRN blocked instigation-heightened aggression but did not change the amount of baseline aggression observed in the standard RI test. It may be that activation of this projection encodes

**Fig. 5 Optogenetic activation of LHb-DRN projection neurons and aggressive behavior. a** Schematic of viral targeting and optic stimulation strategy. AAV2-hSyn-hChR2(H134R)-EYFP virus was injected into the LHb, and an optic fiber was inserted into the DRN. On the test day, optic stimulation was applied to the DRN (473 nm, 20 Hz, 10 ms pulse, 3 mW). **b** Expression of ChR2-EYFP (green) in bilateral LHb. Inset shows an enlarged picture of the LHb (scale bar 200 μm). Blue: DAPI. **c** ChR2-EYFP-expressing LHb projections (green) were observed in the DRN (outlined in white lines). Putative fiber insertion site was indicated as yellow dotted line. Red: Tph2 antibody immunohistochemistry (scale bar 200 μm). **d** c-Fos expression in the DRN after 1 min light stimulation. Cyan: Tph2 antibody, Magenta: c-Fos antibody immunohistochemistry (scale bar 200 μm). Inset shows enlarged picture of c-Fos expression in Tph2+ neurons (filled arrowhead) and Tph2- cells (arrowhead without fill). Light stimulation increased c-Fos expression in both Tph2+ 5-HT neurons (**e** unpaired t test (two-sided), OFF $n = 6$, ON $n = 8$ biologically independent animals, $t(12) = 4.842$, $p = 0.0004$) and Tph2- non-5-HT neurons (**f** unpaired t test (two-sided), OFF $n = 6$, ON $n = 8$ biologically independent animals, $t(12) = 4.297$, $p = 0.0010$). **g** Each animal was tested with 4 types of condition; OFF: No light stimulation, ON: Light stimulation during aggressive encounter, ON/OFF: Light stimulation 1 min before aggressive encounter, but no light stimulation during aggressive encounter, ON/ON: Light stimulation both 1 min before aggressive encounter and during the aggressive encounter. **h** Timeline of this experiment. Aggression test was conducted once a day, and OFF sessions were conducted between any type of stimulation sessions. **i** Duration of total aggressive behaviors in each session. Black line indicates average data of 9 animals (Mean ± SEM), and gray line indicates each individual's data. **j** Summarized data for the duration of total aggressive behavior in each condition. Value of OFF session was calculated as average data from day 1, 3, 6, and 8. Both ON and ON/ON sessions showed significantly longer duration of aggressive behavior than the OFF session (one-way repeated measures ANOVA, $n = 9$ biologically independent animals, $F(3,24) = 9.8265$, $p = 0.0002$, post hoc t test with Tukey's multiple comparisons test (two-sided)). **k–n** Detailed behavioral analysis showed that both ON and ON/ON sessions increased aggressive behaviors (one-way repeated measures ANOVA, $n = 9$ biologically independent animals, attack bites: $F(3,24) = 14.320$, $p < 0.0001$ (**k**), sideways treats: $F(3,24) = 6.754$, $p = 0.0018$ (**l**), post hoc t test with Tukey's multiple comparisons test (two-sided)), but not non-aggressive behaviors (**m** locomotion) compared to OFF sessions. Social interaction was increased in ON compared to OFF sessions (**n** one-way repeated measures ANOVA, $n = 9$ biologically independent animals, $F(3,24) = 6.173$, $p = 0.0029$, post hoc t test with Tukey's multiple comparisons test (two-sided)). **o** Timeline of subthreshold social instigation (S-Inst) test. S-Inst: A caged-instigator male was presented for 1 min before 5-min aggressive encounter. ON/OFF: 1 min light stimulation was delivered during S-Inst, but no light stimulation during aggressive encounter. **p** Duration of total aggressive behaviors in each session. Black line indicates average data of 9 animals (Mean ± SEM), and gray line indicates each individual's data. **q** Summarized data for the duration of total aggressive behaviors in each condition. Value of OFF session was calculated as average data from Day 1 and 3. Combination of S-Inst and light stimulation (ON/OFF), but not S-Inst without light stimulation (OFF), showed significant increase of the duration of total aggressive behavior (Friedman test (two-sided), $n = 9$ biologically independent animals, Friedman statistic = 14.00, $p = 0.0002$, post hoc test with Dunn's multiple comparison test). **r** Optogenetic stimulation of LHb-DRN projection (ON/ON) did not induce female-directed aggressive behavior. Each bar represents mean value ± SEM, and gray line indicates each individual's data. $*p < 0.05$, $**p < 0.01$, $***p < 0.001$, $****p < 0.0001$. Source data are provided as a Source Data file.

the mode of aggressive arousal induced by social provocation, which subsequently enhances the level of intermale aggressive behavior.

The LHb has been implicated as an important node of the brain's reward circuity that integrates emotional valence to drive the selection of actions[26]. Recent evidence has illustrated the involvement of habenular nuclei in aggressive behavior in several animal species including humans[27–30]. From these studies, it is clear that the LHb microcircuitry is complex and when engaged by aggression can have varying effects depending on the cell type and downstream projection. For example, it has been shown that the activity of glutamate neurons, which are the predominant cell type in the LHb, is higher in male mice that do not show aggressive behavior during aggressive encounters (non-aggressors) compared to aggressive individuals[28,29], and a small subset of GABAergic LHb neurons is oppositely regulated in non-aggressors compared to aggressors[29]. However, we show here that there is a subset of glutamate LHb neurons projecting specifically to the DRN that, when activated, cause escalation of aggressive behavior of male mice in response to social instigation. These LHb neurons were not activated in the standard RI test in aggressors. In zebrafish, subregions of dorsal habenula (dHb; which corresponds to the medial habenula in mouse) have opposing effects on aggression, where the lateral subregion of the dHL (dHbL) facilitates winner behavior and the medial subregion of dHb (dHbM) enhances loser behaviors[27]. Interestingly, the winner-promoting dHbL neurons project to the dorsal and intermediate interpeduncular nucleus (IPN), which sends afferent projections to the dorsal tegmental area through the DRN. On the other hand, the loser-promoting dHbM projects to the ventral IPN, which then projects to the median raphe nucleus[27]. The mammalian LHb sends major afferent projections to the VTA and rostromedial tegmental area (RMTg) in addition to the DRN, and some studies have shown that LHb neurons respond to stress

differently depending on its projection area[31,32]. Thus, it is likely that depending on the neural projection target, activation of LHb neurons can have different effects on aggressive behavior.

The LHb sends glutamatergic projections to both serotonergic and non-serotonergic neurons in the DRN[18,19]. Indeed, our data confirm that optogenetic stimulation of LHb-DRN neurons with ChR2 activate both serotonergic and non-serotonergic neurons. However, we found that optogenetic inhibition of 5-HT neurons in the DRN did not have any effect on aggressive behavior or instigation-heightened aggression, suggesting that non-serotonergic neurons may be responsible for LHb-DRN effects on aggression. Anatomical analysis using AAV1-Cre to label specific projections showed that LHb neurons synapse on DRN neurons that project to the VTA, most of which are non-serotonergic glutamatergic neurons that expresses Vglut3[33]. Functional studies where we optogenetically activate the DRN-VTA projection confirmed that they do in fact increase aggressive behavior. The Vglut3+ DRN neurons have been shown to project directly to VTA dopamine neurons, and optogenetic activation of the DRN-VTA projection induces dopamine release in the nucleus accumbens and increases reward-related behaviors[33,34]. Since optogenetic activation of dopamine neurons in the VTA also increases aggressive behavior in male mice[25], it is likely that LHb-DRN-VTA projections activate VTA dopamine neurons to increase aggressive behavior. Other groups have reported the involvement of non-serotonergic neurons in the DRN on aggressive behavior. Optogenetic suppression of non-serotonergic CaMKIIa-expressing neurons in the DRN reduced the duration of aggressive behavior, and activation of non-serotonergic DRN projections to the medial orbitofrontal cortex increased, while its projection to the medial amygdala decreased, the duration of aggressive behavior[35]. However, in our current tracing studies using AAV1-Cre we did not find visible projections in either brain area, and thus the LHb projection might not have as strong

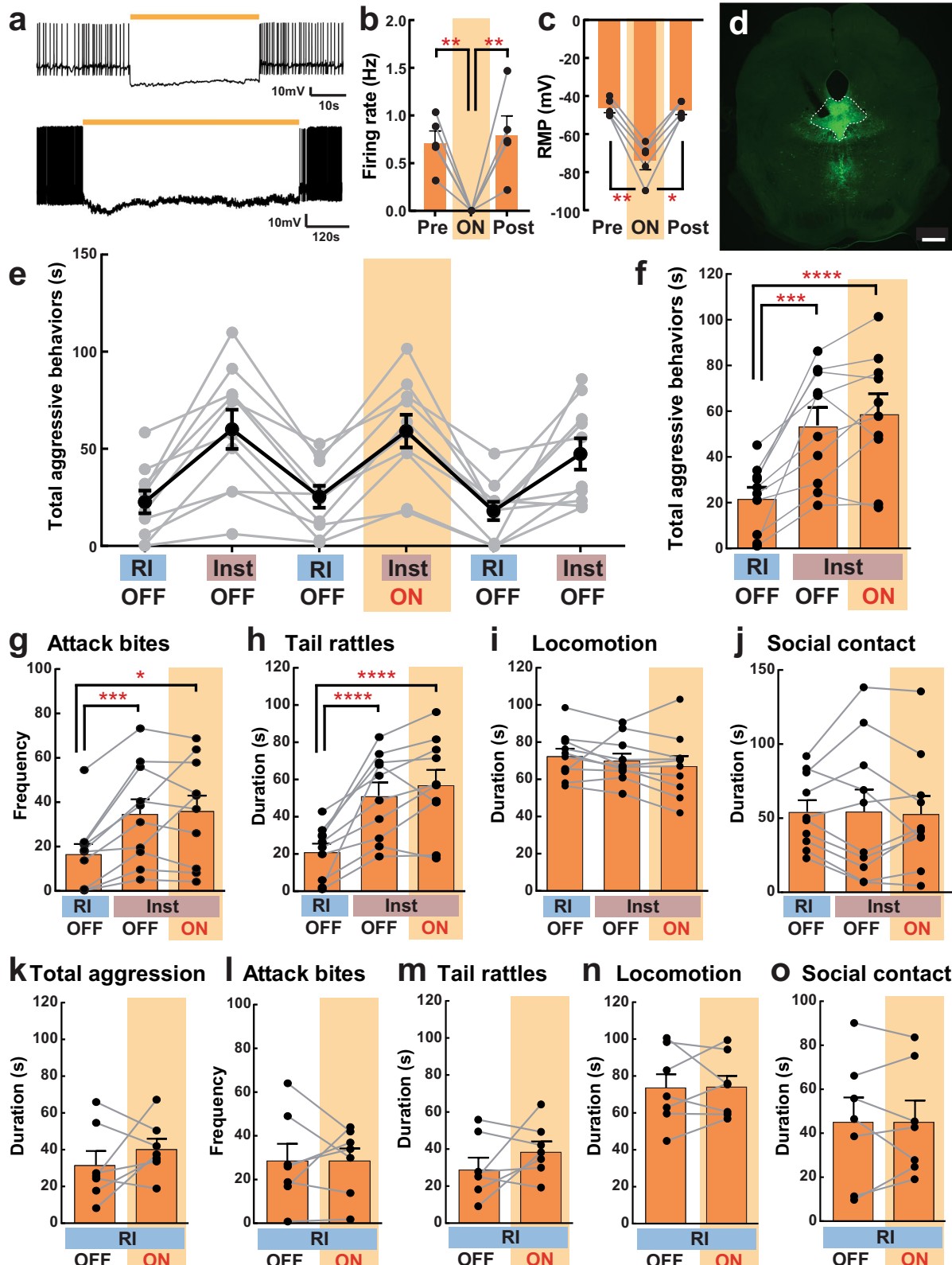

of an effect on those populations of DRN neurons. In addition to the VTA, we found anatomical connections of LHb-DRN neurons that project to the medial mammillary nucleus, caudal and lateral subdivision of interpeduncular nucleus, and LH, as well as feedback connections to the LHb. Future studies will need to examine the roles of these other projections, as well as neural

inputs other than the LHb into the DRN, in mediating aggressive behavior.

Although a large body of evidence indicates that there is an important link between 5-HT and aggressive behavior[9–13], the current study showed that optogenetic inhibition of 5-HT neurons in the DRN did not affect aggressive behavior in male mice.

**Fig. 6 Optogenetic inhibition of DRN 5-HT neurons and instigation-heightened aggression. a** Ex vivo slice patch-clamp recording of EGFP + 5-HT neurons in the DRN of Tph2-tTA::TetO-eArchT-EYFP transgenic mice. Yellow lines indicate yellow light illumination (Top: 1 min, bottom: 10 min). **b**, **c** Summarized data for the spontaneous firing and resting membrane potential (RMP) before (Pre), during (ON), and after (Post) the 1-min light illumination, which showed successful suppression of both firing and RMP (one-way repeated measures ANOVA with Geisser-Greenhouse correction, $n = 5$ biologically independent animals, firing rate: $F(1.486,5.945) = 15.06$, $p = 0.0060$ (**b**), RMP: $F(1.028,4.113) = 70.22$, $p = 0.0010$ (**c**), post hoc t test with Tukey's multiple comparisons tests (two-sided)). **d** Expression of eArchT-EYFP in the DRN and median raphe nucleus. Optic fiber was placed into the DRN (scale bar 500 µm). The timeline of the experiment was exactly the same as in Fig. 3. **e** Duration of total aggressive behaviors in each session. Gray lines show each individual's data ($n = 10$ biologically independent animals) and black line indicates average data (Mean ± SEM). **f** Summarized data for the duration of total aggressive behavior in each condition. The Inst test increased total aggressive behavior compared to RI test in both OFF and ON sessions (one-way repeated measures ANOVA, $n = 10$ biologically independent animals, $F(2,18) = 28.04$, $p < 0.0001$, post hoc t test with Tukey's multiple comparisons tests (two-sided)). **g–j** Detailed behavioral analysis showed that optogenetic inhibition of 5-HT neurons did not block the pro-aggressive effect of social instigation, including attack bites (**g** Friedman test (two-sided), $n = 10$ biologically independent animals, Friedman statistic = 16.80, $p < 0.0001$, post hoc test with Dunn's multiple comparison test (two-sided)), and tail rattles (**h** one-way repeated measures ANOVA, $n = 10$ biologically independent animals, $F(2,18) = 29.20$, $p < 0.0001$, post hoc t test with Tukey's multiple comparison tests (two-sided)), and did not affect non-aggressive behaviors (one-way repeated measures ANOVA, $n = 10$ biologically independent animals, n.s. for both locomotion (**i**) and social contact (**j**)). **k–o** No effect of light stimulation was observed in the RI test on the duration of total aggressive behavior (**k**), frequency of attack bites (**l**), duration of tail rattles (**m**), locomotion (**n**), and social contacts (**o**) (Mann–Whitney test (**k**) or paired t test (**l**, **m**, **n**, **o**), all two-sided, $n = 10$ biologically independent animals, n.s.). Each bar represents mean value ± SEM, and gray line indicates each individual's data. *$p < 0.05$, **$p < .01$, ***$p < 0.001$, ****$p < 0.0001$. Source data are provided as a Source Data file.

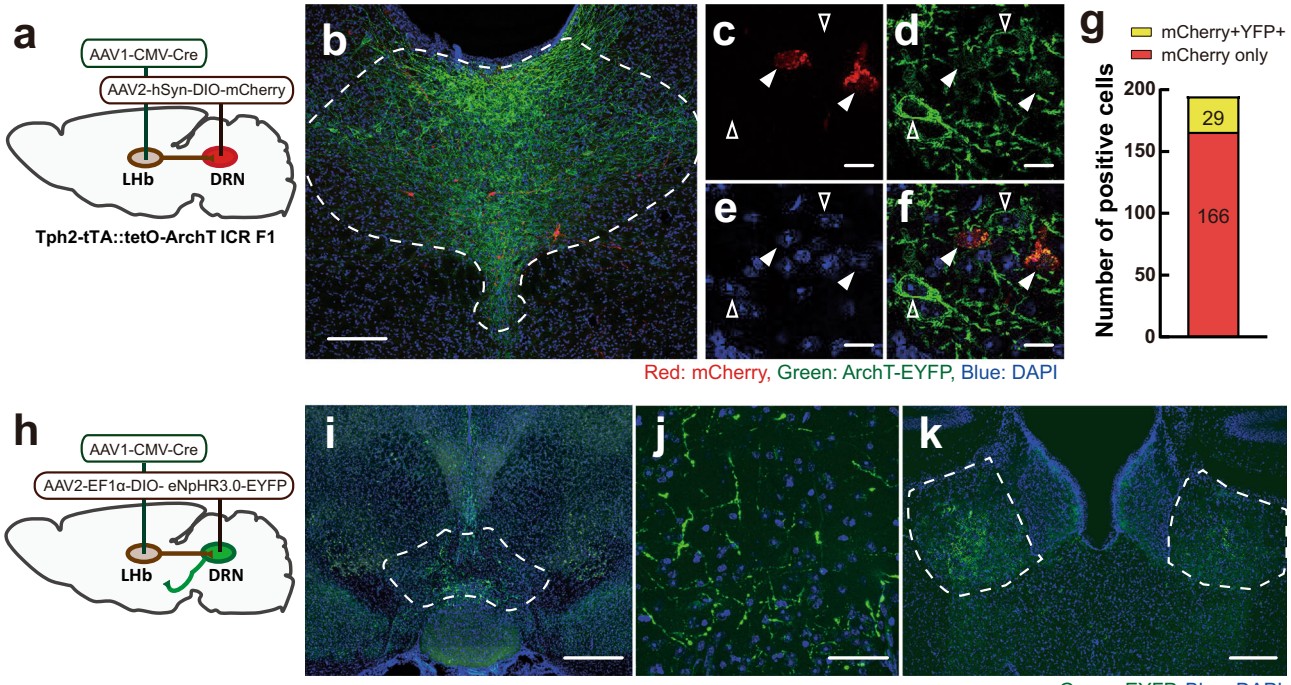

**Fig. 7 LHb-DRN projection mainly targets non-5-HT neurons in the DRN. a** Schematics of AAV1-CMV-Cre injection into unilateral LHb and Cre-dependent AAV2-hSyn-DIO-mCherry injection into the DRN of Tph2-tTA::TetO-eArchT-EYFP transgenic mice. **b** mCherry-expressing cells (red) were observed in the DRN. 5-HT neurons were labeled by eArchT-EYFP (green) (scale bar 200 µm). Enlarged pictures of mCherry (**c**), ArchT-EYFP (**d**), DAPI (**e**), and their merge (**f**) (scale bar 50 µm). Arrowheads indicate ArchT-EYFP-expressing cells (without fill) and ArchT-EYFP- mCherry+ cells (filled). **g** Ratio of mCherry+ cells that colocalized with ArchT-EYFP (yellow, 29 cells (14.9%)) and without ArchT-EYFP (red, 166 cells (85.1%)) in the DRN ($n = 3$ biologically independent animals). **h** Schematics of AAV1-Cre injection into unilateral LHb and Cre-dependent AAV2-EF1α-DIO-eNpHR3.0-EYFP into the DRN to observe projections to forebrain areas. **i–k** EYFP-expressing fibers (green) were observed in the medial part of the VTA (**i** scale bar 500 µm, **j** scale bar 50 µm), and bilateral LHb (**k** scale bar 200 µm). Blue: DAPI. Source data are provided as a Source Data file.

Other groups have also reported that acute optogenetic activation of DRN neurons did not have a strong effect on aggressive behavior, but rather chronic activation of the DRN over 8 days significantly reduced aggressive behavior[36]. Indeed, we observed differences in 5-HT neural responses to social encounters between aggressor and non-aggressor mice, suggesting a link between 5-HT neural reactivity and aggressive traits[37]. One interesting possibility is that chronic dysregulation of the 5-HT system affects the activity of downstream projections such as the VTA to control aggressive behavior. Whether this is the case should be investigated in future studies.

In conclusion, we show that a subpopulation of LHb neurons that project to the DRN promotes aggressive arousal, and increases social instigation-induced escalation of aggressive behavior in male mice without affecting species-typical aggression. Our data indicate that VTA-projecting non-serotonergic

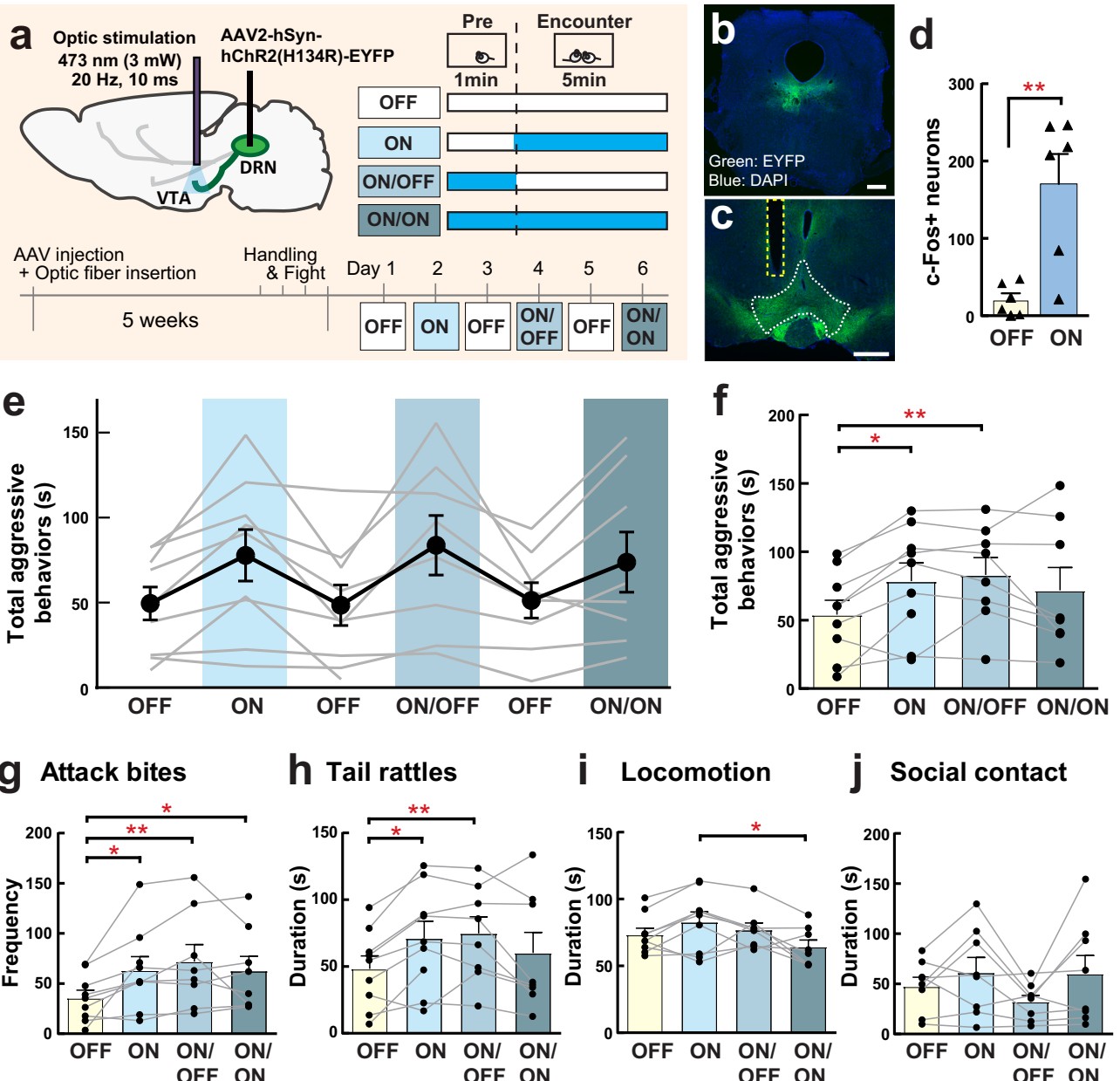

**Fig. 8 Optogenetic activation of DRN-VTA projection neurons increases aggressive behavior. a** Schematics and timeline of this experiment. AAV2-hSyn-ChR2-EYFP was injected into the DRN, and the optic fiber was inserted into the VTA. Stimulation schematics were same as in Fig. 5. Expression of ChR2-EYFP (green) in the DRN (**b** scale bar 500 μm) and their projections in the VTA (**c** scale bar 500 μm). Blue: DAPI. **d** Light stimulation increased c-Fos expression in the VTA after 1 min light stimulation (unpaired t test (two-sided), OFF $n = 6$, ON $n = 6$ biologically independent animals, $t(10) = 3.796$, $p = 0.0035$). **e** Duration of total aggressive behaviors in each session. Black line indicates average data of 9 animals (Mean±SEM), and gray line indicates each individual's data. **f** Summarized data for the duration of total aggressive behavior in each condition. Value of OFF session was calculated as average data from Day 1, 3, and 5. Both ON and ON/OFF sessions showed significantly longer duration of aggressive behavior compared to the OFF session (mixed-effects analysis (two-sided), $n = 9$ biologically independent animals, $F(1.977,14.50) = 5.133$, $p = 0.0209$, post hoc $t$ test with Tukey's multiple comparisons test (two-sided)). Detailed behavioral analysis showed that all stimulation schematics (ON, ON/OFF, ON/ON) increased attack bites compared to OFF session (**g** mixed-effects analysis (two-sided), $n = 9$ biologically independent animals, $F(3,22) = 6.268$, $p = 0.0031$, post hoc $t$ test with Tukey's multiple comparisons test (two-sided)), whereas only ON and ON/OFF sessions showed significant increases in the duration of tail rattles compared to OFF session (**h** mixed-effects analysis (two-sided), $n = 9$ biologically independent animals, $F(1.949,14.30) = 5.669$, $p = 0.0159$, post hoc $t$ test with Tukey's multiple comparisons test (two-sided)). By contrast, ON/ON session showed a significant reduction of locomotion compared to ON session (**i**; mixed-effects analysis (two-sided), $n = 9$ biologically independent animals, $F(1.977,14.50) = 5.133$, $p = 0.0209$, post hoc $t$ test with Tukey's multiple comparisons test (two-sided)). No effect of light stimulation was observed in social contact (**j** Friedman test (two-sided), $n = 9$ biologically independent animals, n.s.). Each bar represents mean value ± SEM, and gray line indicates each individual's data. *$p < .05$, **$p < .01$. Source data are provided as a Source Data file.

DRN neurons might be a possible target for LHb-DRN gluta-matergic inputs to increase aggression. Therefore, the LHb-DRN-VTA pathway might be involved in socially provoked anger or violence.

## Methods

**Animals**. Sexually naïve adult ICR (CD-1) males were purchased from Charles River Laboratories Japan. These animals were 7–9 weeks old upon the time of arrival to our facility, and 10–15 weeks at the time of behavioral experiments. For 5-HT neuron optogenetics, Tph2-tTA::TetO-eArchT-EYFP transgenic male mice were used. Due to low aggressive behaviors in the original genetic background of this transgenic strain, F1 offspring were made by crossing it with CD-1 females after confirming that each animal had both Tph2-tTA and TetO-ArchT-EYFP alleles by genotyping. These animals were 11–16 weeks old at the time of behavioral experiments. For stimulus (intruders), adult ICR/Jcl males were used that were originally derived from Japan CLEA and were bred and maintained in animal facility of University of Tsukuba. These intruder animals were olfactory bulbec-tomized at least 1 week before the experiment to reduce intruders' aggressive behavior but were still able to produce aggression-promoting male odors due to intact testis[38]. Only for the LHb-DRN ChR2 (Supplementary Fig. 6) and DRN-VTA ChR2 (Supplementary Fig. 12) optogenetics experiments in a different facility, sexually experienced CD-1 mice that were ~4 months of age (Charles River Laboratory) were used as test mice, and male BALB/cByJ mice (more than 10 weeks old, The Jackson Laboratory) were used as intruders. To examine female-directed behavior, C57BL/6J females (10–12 weeks old, Japan CLEA) were used.

Resident ICR males were housed individually in standard mouse cages with corn cob bedding material, and intruder animals were group housed (3 to 5 males per cage) in the same standard mouse cages with corn cob bedding material throughout the experiment. All animals were maintained in the animal rooms with controlled temperature ($23 \pm 2$ °C, average humidity; 50%) on a 12 h light/dark cycle with ad libitum access to food and water. Experimental procedures were performed in accordance with the National Institutes of Health Guide for Care and Use of Laboratory animals, the Animal Care and Use Committee at the University of Tsukuba (approval number 20-341, 20-342, 21-304, 21-422), and the Icahn School of Medicine at Mount Sinai (ISMMS) Animal Care and Use Committee (approval number LA10-00266). Behavioral experiments were conducted during the dark cycle under red light except for the LHb-DRN ChR2 experiment (Supplementary Fig. 6) and DRN-VTA ChR2 experiment (Supplementary Fig. 12) in other facility, which was conducted during the light cycle. There was no difference between light and dark cycles in the effect of optogenetic stimulation on either LHb-DRN or DRN-VTA projection on aggressive behavior.

**Aggression test (Resident-intruder test and social instigation test)**. To measure species-typical aggressive behavior and heightened aggression by social provocation, we conducted a resident-intruder (RI) test and a social instigation (Inst) test. All aggression tests were conducted in the resident male's homecage. Right before the experiment, the cage top of their homecage was substituted with clear acrylic top. In the RI test, an intruder male was introduced into the homecage of ICR resident male, and their behaviors were videotaped for 5 min from the side of cage. In the Inst test, an adult instigator male was placed in a circular cylinder with holes (φ7 cm × 12 cm (height) acrylic cylinder with 5 mm holes) and placed in the middle of the homecage of ICR resident male for 5 min. During this instigation period, resident males can see, smell, and feel the existence of instigator male but cannot physically attack it, thus it is considered to induce aggressive arousal[4]. Five minutes later, the cylinder with the instigator was removed from the homecage, and an intruder male (which was a different animal than the instigator) was placed in the homecage of the resident male and their behaviors were recorded for 5 min. We also conducted a modified version of the Inst test termed subthreshold social instigation test (S-Inst). In this test, a caged-instigator male was presented for 1 min before the 5-min aggressive encounter. After the aggression test, the intruder male was returned to its homecage, and the regular cage top with food and water was returned to resident's homecage. One resident male experienced several aggressive encounters in most of the experiments (see detail in the following section), and they always encountered with the same intruder throughout the experiment in both RI and Inst tests. By contrast, novel instigator was used for each Inst test.

Detailed behavioral analysis during aggression tests were scored from the recorded video by a well-trained observer, who was blind to experimental conditions, to quantify aggressive behaviors (attack bites, sideways threats, tail rattles, pursuits) and non-aggressive behaviors (locomotion, rearing, grooming, and social contacts)[39,40] using tanaMove v0.01, a free software established by A. Tanave (http://www.mgrl-lab.jp/tanaMove.html). The frequency of attack bites, latency to the first bite, and durations of other behaviors were used for the analysis. In addition, the duration of total aggressive behavior was calculated by occurrence of either attack bites, sideways threats, tail rattles, or pursuits.

**RetroBeads injection and c-Fos analysis**. Stereotaxic surgeries were conducted under isoflurane inhalation anesthesia. RetroBeads (RetroBeads™ Red, Lumafluor Inc) were injected into the DRN (AP −4.6 mm; ML +1.5 mm; DV −4.0 mm to bregma, angled 26° to the vertical)[41] by using 33-gauge needle attached to

Hamilton syringe (800 series RN, Hamilton Company). Two hundred nanoliters of Retrobeads were injected at the speed of 50 nL/30 s, and the needle was left there for 10 min after the injection to let the beads diffuse around the target area. After stitching the outer skin of the scalp, animals were returned to homecage and housed individually for 10 days.

In the test, resident males were assigned to one of three test groups: RI group ($n = 10$), Inst group ($n = 12$), or control (Cont) group ($n = 10$). After histological verification, the number of animals used for c-fos expression analysis was as follows: RI group $n = 8$, Inst group $n = 10$, Cont group $n = 8$. As described above (Aggression test), resident males of the RI group experienced a 5 min RI test, and resident males of the Inst group experienced 5 min social instigation followed by a 5 min RI test. At the end of the aggression test, an intruder male was removed and resident males stayed in homecage without any disturbance for 85 min until perfusion. Males of the Cont group were kept in their homecage without any aggressive encounter. All animals were deeply anesthetized with ketamine (100 mg/kg) and xylazine (10 mg/kg) mix. Animals were first perfused with 40 mL of ice-cold 0.1 M phosphate-buffered saline (PBS) and then 40 mL of ice-cold 4% paraformaldehyde (PFA) at the speed of 4 mL/min using a peristaltic pump (Dynamax RP-1, Rainin). Their brains were removed and post-fixed with 4% PFA overnight at 4 °C, and then placed in 30% sucrose in PBS at 4 °C for 2 nights. Brains were then frozen in isopentane on dryice, and kept in −80 °C until immunohistochemistry tissue processing.

**DREADD-mediated inhibition of the LHb-DRN projection**
*Stereotaxic surgery*. Stereotaxic surgeries were conducted under isoflurane inhalation anesthesia. We assigned half of animals to the hM4D(Gi) DREADD group ($n = 17$) and the other half to the control EYFP group ($n = 17$). Among those, 9 animals that did not show instigation-heightened aggression with saline (SAL) injection, and 1 animal without M4 expression were excluded from the analysis. The final number of animals used in this analysis was $n = 11$ for hM4D DREADD and $n = 11$ for EYFP control. All animals were injected with retrograde AAVretro-EF1α-mCherry-IRES-Cre (the plasmid was a gift from Karl Deisseroth, Addgene plasmid # 55632) into the DRN (AP −4.6 mm; ML +1.5 mm; DV −4.0 mm to bregma, angled 26° to the vertical)[41]. Then, a Cre-dependent AAV was bilaterally injected into the LHb (AP −1.7 mm; ML ±0.9 mm; DV −2.7 mm to bregma, angled 10° to the vertical)[41]. Males of the hM4D DREADD group received AAV2-CAG-Flex-rev-hM4D-2A-EGFP (Addgene plasmid # 52536)[42] and males of the control EYFP group received AAV2-EF1α-DIO-ChR2(H134R)-EYFP (the plasmid was a gift from Karl Deisseroth, Addgene plasmid # 20298). Two weeks after the AAV injection, test animals were handled and then an intruder was introduced into the homecage for 5 min to let them habituate to the test condition and aggressive encounter. This habituation session was repeated once a day until resident males showed aggressive behavior for at least two consecutive days.

*Behavior experiment*. Resident males were tested in both the RI test and the Inst test alternately, and each test was conducted once per 2 days. Animals were intraperitoneally (i.p.) injected with SAL or 1 mg/kg Clozapine-N-oxide dissolved in saline (CNO, Enzo Life Sciences). First CNO injection was 4 weeks after the AAV injection. In both the RI test and the Inst test, an intruder male was introduced into the resident's homecage 35 min after i.p. injection and their aggressive behavior was observed for 5 min. Only in the Inst test, an instigator male in a protective cage was presented in resident's homecage for 5 min prior to the aggressive encounter. On days 1, 5, and 9, animals received SAL injections and then were tested in the RI test (SAL + RI). On days 3 and 11, animals received SAL injections and were tested in the Inst test (SAL + Inst). On day 7, animals received CNO and then were tested the Inst test (CNO + Inst) (Fig. 3a). Three days after the last Inst test, the effect of CNO was tested in the RI test, and a SAL + RI session was followed by a session with a CNO injection (CNO + RI) in two-day intervals. At the end of the experiment, their brains were obtained for histological verification as described above.

**Optogenetic experiments**
*Stereotaxic surgery*. Stereotaxic surgeries were conducted under isoflurane inhalation anesthesia. Either AAV2-hSyn-hChR2(H134R)-EYFP (Addgene plasmid # 26973, UNC Vector Core), AAV2-hSyn-eNpHR3.0-EYFP (Addgene plasmid # 26972), AAV2-hSyn1-EYFP (Addgene plasmid # 117382), or AAV2-hSyn-mCherry (UNC Vector Core) [all of the plasmids were gifts from Karl Deisseroth] was injected at a volume of either 300 nL/side into bilateral LHb (for LHb-DRN: AP −1.7 mm; ML ±0.9 mm; DV −2.7 mm to bregma, angled 10° to the vertical)[41] or 500 nL into the DRN (for DRN-VTA: AP −4.6 mm; ML +1.5 mm; DV −4.0 mm to bregma, angled 26° to the vertical)[41], and the needles were left in place for 10 min after the injection to let the virus to diffuse in the area. At the same time, 250 µm fiber-optic cannula (handmade with ceramic ferrule (CFLC270; Thorlabs) with 250 µm polymethyl methacrylate fiber (PJS-FB250; Toray Industries, Inc)) were stereotaxically inserted into either the DRN (for ChR2 experiment: AP −4.6 mm; ML 0 mm; DV −2.3 mm to bregma without angle, for eNpHR3.0 and Tph2-tTA::tetO-eArchT-EYFP experiments: AP −4.6 mm; ML +1.5 mm; DV −3.3 mm to bregma, angled 26° to the vertical)[41] or the VTA (AP −3.2 mm; ML +0.9 mm; DV −4.0 mm to bregma, angled 7° to the vertical)[41].

*Optogenetic stimulation and inhibition*. One month after optic fiber insertion, test animals were held by hand and their optic cannula were connected to a patch cord. The tethered mice were then placed in the test cage (19.2 × 29 × 30 cm, with bedding materials moved from their homecage) once per day to let them habituate to the test condition. During this habituation period, an intruder male was introduced into the test cage with the resident male for 5 min. This handling/habituation session was repeated daily until resident males showed aggressive behavior for two consecutive days. Behavior tests with optical stimulation was started at least five weeks after AAV injection for optimal expression of the opsin in the projection terminal[33]. In the optogenetic experiments, the optic cannula of the resident male was connected to a patch cord attached to a rotary joint (FRJ_1×1_FC-FC; Doric Lenses Inc) and then connected to a light source with 473 nm blue laser and 593 nm yellow laser (COME2-LY-1; Lucir Inc) which was modulated by a schedule stimulator (Lucir Inc). Right after connection of the patch cord, the animal was placed in the test cage. For ChR2 stimulation, optic stimulation consisted of 20 Hz frequency of 10 ms blue light pulses with the intensity of approximately 3 mW at the tip of fiber-optic cannula. For eNpHR3.0 experiment, continuous yellow light was applied with the intensity of about 3 mW at the tip of fiber-optic cannula.

In eNpHR3.0 experiments for the LHb-DRN ($n = 9$), resident males were tested in both the RI test and the Inst test alternately, and either test was conducted once per 2 days (Fig. 4a). Light stimulation was delivered at the 2nd Inst session, throughout the social instigation and aggressive encounter (5 min + 5 min = 10 min), as well as during the 5th RI session during aggressive encounter only (5 min). At the end of the experiment, brains were obtained for histological verification as described above. For terminal inhibition of the LHb-DRN projection, eNpHR3.0 was used based on previous studies[28,43].

In the LHb-DRN ChR2 experiment ($n = 9$), an aggressive behavior test was conducted once a day with different light stimulation patterns (Fig. 5g), and all animals were tested using the same order of stimulation schemes. On Day 2, light stimulation was started 5 s before the introduction of intruder male, and continued until the end of the 5-min aggressive encounter (ON session). On Day 4, light stimulation was delivered for 1-min before the entry of intruder into test cage, and no light stimulation was delivered during the 5-minute aggressive encounter (ON/OFF session). On Day 7, light stimulation was initiated 1 min before the entry of intruder, and it was continued until the end of the aggressive encounter (ON/ON session). On days 1, 3, 6, and 8, the RI test was conducted without any light stimulation (OFF session). Among 10 males tested in this study, one male did not show any aggressive behavior throughout the experiment and thus was excluded from the analysis. One day after the last OFF session, animals received a 1-minute stimulation (20 Hz, 10 ms blue light pulses, 3 mW) without any aggressive encounter for c-Fos expression analysis. Ninety minutes after the optic stimulation, animals were deeply anesthetized with ketamine (100 mg/kg) and xylazine (10 mg/kg) mix, perfused with ice-cold PBS and then 4% PFA, and their brains were post-fixed with 4% PFA overnight, 30% sucrose for 2 nights, and kept in −80 °C until immunohistochemistry analysis. In a separate group of animals, we stimulated LHb-DRN with ChR2 ($n = 9$) during a subthreshold social instigation test (S-Inst). In this test, a caged-instigator male was presented for 1 min before the 5-min aggressive encounter in RI. On Day 2, S-Inst test was conducted without light stimulation (S-Inst+OFF). On Day 4, light stimulation was delivered for 1-min combined with the presentation of an instigator (S-Inst) right before the entry of an intruder into test cage, and no light stimulation was delivered during the 5-min aggressive encounter (S-Inst+ON/OFF). On Days 1 and 3, the RI test was conducted without any light stimulation (RI + OFF). A few days after completing S-Inst tests, we tested the effect of LHb-DRN ChR2 stimulation on female-directed behavior. A C57BL/6J female was introduced to the test cage, and the behavior of the test males were recorded for 5 min without (Day 1) and with (Day 2) light stimulation. In addition, the effect of LHb-DRN stimulation on spontaneous locomotor activity was examined in their homecage. The mouse was connected to an optic fiber and placed in the homecage without the lid, and the behavior was recorded for 6 min. One-minute light stimulation (20 Hz, 10 ms blue light pulses, 3 mW) was delivered in an alternating pattern starting with 1 min OFF and then 1 min ON. Homecage activity was measured by tracking the position of animal using DeepLabCut from the recorded video. Among 10 males tested in this study, one male with low ChR2 expression was excluded from the analysis.

In the DRN-VTA ChR2 experiment (ChR2 $n = 9$, EYFP control $n = 8$), the same experimental procedure was used as the LHb-DRN ChR2 experiment (Fig. 8a). Among 28 males tested in this study, 11 males without ChR2 expression was excluded from the analysis.

*ArchT inhibition of DRN 5-HT neuron*. Tph2-tTA::TetO-eArchT-EYFP transgenic mice were used to inhibit 5-HT neuron activity during aggressive encounters. Because we aimed to inhibit 5-HT neurons throughout social instigation, we chose to use ArchT which is a better opsin for long-term neural inhibition[44,45]. Due to low aggressive behaviors in the original genetic background of this transgenic line, F1 offspring were generated by crossing this line with ICR/CD-1 females. All animals were genotyped for Tph2-tTA and TetO-ArchT-EYFP alleles, and animals that possessed both Tph2-tTA and TetO-ArchT-EYFP alleles were used in this study. Because of the low level of aggression of this transgenic line, we screened animals with 5 min RI tests for 3 days before the experiment. Only animals that showed aggressive behavior in any day of this screening were used ($n = 10$). Experimental procedure for this behavioral experiment was same as LHb-DRN NpHR3.0 experiment (Fig. 4) except that the habituation session started one week after the surgery.

**Anterograde labeling of DRN neurons that receive projections from the LHb**. Stereotaxic surgery was conducted under isoflurane inhalation anesthesia. AAV1-CMV-Cre was injected at a volume of 300 nl/side into unilateral LHb (AP −1.7 mm; ML −0.9 mm; DV −2.7 mm to bregma, angled 10° to the vertical)[41], and 500 nl of either AAV2-hSyn-DIO-mCherry (Addgene plasmid # 50459) or AAV2-EF1α-DIO-eNpHR3.0-EYFP (Addgene plasmid # 26966, both plasmids were gifts from Karl Deisseroth) was injected into the DRN (AP −4.6 mm; ML +1.5 mm; DV −4.0 mm to bregma, angled 26° to the vertica l)[41]. Five weeks after the injection, animals were deeply anesthetized with a mixture of ketamine (100 mg/kg) and xylazine (10 mg/kg), perfused with ice-cold PBS and then 4% PFA, and their brains were post-fixed with 4% PFA overnight, 30% sucrose for 2 nights. Brains were kept in −80 °C until immunohistochemistry tissue processing.

**Histology and immunohistochemistry**. Coronal brain slices were made using a cryostat (thickness: 30 μm for immunohistochemistry and 60 μm for histological verification). Free-floating sections were first washed with PBS and then incubated with a blocking solution (10% Block ACE (DS Pharma Biomedical) + 0.3% Triton-X100 in PBS) at room temperature (RT). Then sections were incubated with primary antibody in blocking solution (5% Block ACE + 0.3% Triton-X100) for 1 night at 4 °C. For RetroBeads c-Fos analysis, 1:2000 anti-rabbit c-Fos antibody (Abcam, ab190289, lot:GR304825-3). For optogenetics and DREADD experiments, 1:6000 anti-goat GFP antibody (Abcam, ab6673, lot: GR3371856-3) was used to examine ChR2-expressing and M4 DREADD-expressing cells. Separate sets of sections were analyzed for c-Fos expression in DRN serotonergic neurons with 1:2000 anti-rabbit c-Fos (Abcam, ab190289, lot:GR304825-3) and 1:1000 anti-goat Tph2 antibody (Abcam, ab121013, lot: GR3271744-12). For the AAV1-Cre labeling experiment, 1:2000 anti-chicken mCherry antibody (Abcam, ab205402, lot: GR3271744-12), 1:3000 anti-goat GFP antibody (Abcam, ab6673, lot: GR3371856-3), or 1:200 anti-rabbit GFP antibody (Abcam, ab290, lot: 514983), combined with 1:1000 anti-goat Tph2 antibody (Abcam, ab121013, lot:GR3271744-12) or 1:1000 anti-mouse TH(F-11) antibody (Santa Cruz Biotechnology, sc-25269, lot: GR176206-49) were used. After washing with PBS, sections were incubated with secondary antibodies (1:1250 anti-rabbit Alexa488, anti-rabbit Alexa594, anti-goat Alexa488, anti-goat Alexa680, or anti-chicken Alexa594, Jackson ImmunoResearch) for 2 h at RT. After washing with PBS, the sections were incubated with DAPI for 15 min and then coverslipped with Fluoromount™ (Cosmo Bio) mounting medium.

Microscopic images were obtained by all-in-one fluorescence microscope (BZ-X710, Keyence Cooperation, Osaka, Japan). 10 μm-thick z-stack Images of the lateral habenula (LHb) and the dorsal raphe nucleus (DRN) were acquired at 20× magnification for the c-Fos expression analysis. For the histological verification of viral expression and the fiber insertion site, images were acquired at 4× or 10× magnification.

The number of both c-Fos+ neurons and RetroBead-labeled neurons were manually counted using Photoshop CC 2018 (Ver. 19.0, Adobe Inc.). Both right and left sides of the LHb from 6 consecutive slices with a 90 μm interval (approximately AP −1.34 mm to −1.94 mm from bregma)[41] were analyzed and the average number of c-Fos+ and RetroBead-labeled cells in the LHb per slice was used for statistical analysis. The number of c-Fos+ cells and their colocalization with 5-HT+ cells in the DRN were also manually counted from 8 consecutive slices (approximately AP −4.24 mm to −4.96 mm from bregma)[41]. Average numbers of c-Fos+ cells and c-Fos+ cells colocalized with 5-HT+ cells per a slice were used for statistical analysis. Similarly, analysis of AAV1-Cre anterograde labeling was manually conducted using Photoshop. Total number of mCherry+ cells (or EYFP+ cells for Tph2 and TH analyses), as well as the number of mCherry+ cells that were colocalized with ArchT-EYFP or Tph2+ and TH+ cells from 8 consecutive slices with a 90 μm interval (approximately AP −4.24 mm to −4.96 mm from bregma)[41] were analyzed. Although ArchT-EYFP expression was observed the membranes of cell bodies, dendrites and axons, we defined the ArchT-EYFP+ cells when mCherry+ cells were roundly surrounded by EYFP.

**In situ hybridization with retrograde or anterograde labeling**. *Vglut2* expression in the DRN-projecting LHb neurons was examined using samples used from "RetroBeads injection and c-Fos analysis", and *Vglut3* expression was examined in the DRN samples used for "AAV1-Cre injection to label DRN neurons that receive projections from the LHb". Probe sequences were reported previously (*Vglut2*[46], and *Vglut3* (RP_050725_03_G04, Allan mouse brain atlas)).

Coronal brain slices were made using a cryostat (thickness: 30 μm) and the slices were kept in anti-freeze solution (30% ethylene glycol, 30% glycerol in 0.02 M Tris-buffered saline (TBS)) at −20 °C until in situ hybridization tissue processing. The sections were washed with PBS with 0.1% Tween-20 (PBST) for 10 min three times and hybridized with digoxigenin (DIG)-labeled RNA probe (1 μg/ml) in hybridization solution (50% formamide, 5× SSC pH 4.5, 1% SDS, 50 μg/ml heparin, 50 μg/ml yeast RNA) in 2 mL collection tube at 65 °C for overnight. On the following day, the slices were washed with Washing Buffer1 (50% formamide, 5 × SSC pH 4.5, 1% SDS) at 65 °C for 30 min, then Washing Buffer2 (50% formamide, 2 × SSC pH 4.5) at 65 °C for 30 min three times. The slices were washed with TBS with 0.1% Tween-20 (TBST) for 5 min three times, then treated with TBST containing 0.3% $H_2O_2$ and 5% dimethyl sulfoxide for 30 min to quench endogenous peroxidase activity. The slices were again washed with TBST 5 min three times. After blocking with 0.5% blocking reagent (Roche Diagnostics) in TBST for 1 h at room temperature, the slices were incubated with primary antibodies

in 0.5% blocking reagent in TBST at 4 °C for two nights. On the fourth day, the slices were washed with TBST for 20 min three times, incubated with Biotin Tyramide (1:50 in Amplification diluent, PerkinElmer) in the presence of 0.0015% $H_2O_2$ at room temperature for 10 min. For signal detection, after washing with TBST 5 min three times, the slices were incubated with Alexa Fluor 488- or Alexa Fluor 568-conjugated Streptavidin (S11223/S11226, 1:200, Molecular Probes) in TBST at room temperature for 30 min. The slices were washed with TBST for 5 min three times, and incubated with a secondary antibody in 0.5% blocking reagent in TBST at room temperature for 2 h. After washing with TBST 20 min three times, the slices were counterstained with 1 µg/ml DAPI (Wako Pure Chemical Industries) in TBST for 15 min. The slices were washed with TBST for 5 min three times, rinsed with 50 mM Tris-HCl pH 7.4, then mounted on MAS-coated slide glasses (Matsunami Glass Industry) with Fluoromount-G (Southern Biotech). The primary antibody was a horse-radish peroxidase-conjugated anti-DIG antibody (1:100, Roche Diagnostics, 11207733910, lot: 35698000) and anti-GFP antibody (1:500, Molecular Probes, A11122, lot: 1232939). The secondary antibody was an Alexa Fluor Plus 488 conjugated goat anti-rabbit IgG (1:500, Invitrogen, A32731, lot: RJ243417).

A series of z-stack fluorescence images was acquired using a laser scanning confocal microscope, the LSM 700 with a 20× objective lens (Carl Zeiss) and saved as a Carl Zeiss Image data (CZI) file. The pixel size and the optical slice thickness were 0.625 µm × 0.625 µm and 4.6 µm, respectively. For colocalization studies, the CZI files were transformed into 3-D images using Imaris software (Ver. 9.2.1, Bitplane Inc.). The 3-D image was displayed in the Surpass View and the colocalization was analyzed. The number of RetroBead-labeled neurons, in which the RetroBeads signals were localized around the nuclei, were counted in the LHb. Among the RetroBead-labeled neurons, the number of the *Vglut2*-expressing neurons were counted. Finally, the ratio of the double-positive neurons to the RetroBead-labeled neurons was calculated. For anterograde labeling using AAV1-Cre, the number of the EYFP-labeled neurons located within the *Vglut3+* area were counted in the DRN. Among the EYFP-labeled neurons, the number of *Vglut3+* neurons were counted. Finally, the ratio of double-positive neurons to EYFP-labeled neurons was counted.

**Ex vivo slice recordings of LHb-DRN neurons.** ICR (CD-1) male mice (3–5 months old) were injected in the DRN (From bregma: AP −4.6 mm; ML +1.5 mm; DV −4.0 mm, with a 26° angle to the vertical) with a 500 nL of retrograde AAV-hSyn-EGFP (Addgene plasmid # 50465). Four-to-six weeks after surgeries, mice underwent either (1) a RI test (5 min encounter in their homecage, *n* = 5) or (2) an Inst test (5 min social instigation followed by a 5 min RI encounter, *n* = 5). Thirty-to sixty minutes after the behavioral test, mice were deeply anesthetized with isoflurane and decapitated. The brain was rapidly removed and chilled in cutting artificial cerebrospinal fluid (aCSF) containing (in mM): N-methyl-D-glucamine 93, HCl 93, KCl 2.5, $NaH_2PO_4$ 1.2, $NaHCO_3$ 30, HEPES 20, glucose 25, sodium ascorbate 5, thiourea 2, sodium pyruvate 3, $MgSO_4$ 10, and $CaCl_2$ 0.5, pH 7.4. The brain was embedded in 2% agarose and coronal slices (200 µm thick) were made using a Compresstome (Precisionary Instruments). Brain slices were allowed to recover at 33 ± 1 °C in aCSF solution for 30 min and thereafter at room temperature in holding aCSF, containing (in mM): NaCl 92, KCl 2.5, $NaH_2PO_4$ 1.2, $NaHCO_3$ 30, HEPES 20, glucose 25, sodium ascorbate 5, thiourea 2, sodium pyruvate 3, $MgSO_4$, and $CaCl_2$ 2, pH 7.4. After at least 1 h of recovery, the slices were transferred to a submersion recording chamber and continuously perfused (2–4 mL/min) with aCSF containing (in mM): NaCl 124, KCl 2.5, $NaH_2PO_4$ 1.2, $NaHCO_3$ 24, HEPES 5, glucose 12.5, $MgSO_4$ 2, and $CaCl_2$ 2, pH 7.4. All the solutions were continuously bubbled with 95% $O_2$/5% $CO_2$. GFP-expressing LHb neurons were visually identified with infrared differential contrast upright microscope (BX51; Olympus). Whole-cell patch-clamp recordings were performed at room temperature using a Multiclamp700A amplifier (Molecular Devices). Recording electrodes (3–5 MΩ) pulled from borosilicate glass were filled with solution containing (in mM): κ-gluconate 115, HEPES 10, KCl 20, MgATP 5, MgCl2 1.5, $Na_2GTP$ 0.5, Na-phosphocreatine 10, and EGTA 2, pH 7.25. Data acquisition (filtered at 10 kHz and digitized at 10 kHz) and analysis was performed with pClamp 11 software (Molecular Devices). Baseline firing rate for the spontaneously firing neurons were recorded for 2–3 min after being allowed to stabilize for 3 min after breakthrough in *I* = 0 mode. Only cells with stable input resistances were included in the analysis (3–6 neurons were used from one mouse). Recording and analysis were conducted blind to the experimental conditions.

**Ex vivo slice recordings of 5-HT neurons in the DRN.** Male and female Tph2-tTA::tetO-eArchT-EYFP bigenic mice (6 weeks old) were used for patch-clamp recordings. Mice were decapitated under deep anesthesia with isoflurane. Brains were extracted and cooled in ice-cold cutting solution containing: 87 mM NaCl, 75 mM Sucrose, 25 mM $NaHCO_3$, 10 mM D (+)-glucose, 7 mM $MgCl_2$, 2.5 mM KCl, 1.25 mM $NaH_2PO_4$ and 0.5 mM $CaCl_2$ bubbled with $O_2$ (95%) and $CO_2$ (5%). Coronal brain slices (250-µm thickness), including the DRN, were prepared with a vibratome (VT1200S, Leica) and maintained for 1 h at room temperature in artificial cerebrospinal fluid (ACSF) containing: 125 mM NaCl, 1.25 mM $NaH_2PO_4$, 26 mM $NaHCO_3$, 10 mM D (+)-glucose, 2.5 mM KCl, 2 mM $CaCl_2$ and 1 mM $MgSO_4$ bubbled with $O_2$ (95%) and $CO_2$ (5%). The electrodes (8–13 MΩ) were filled with an internal solution containing: 130 mM K-

gluconate, 10 mM or 3 mM KCl, 0.05 mM $CaCl_2$, 0.1 mM $MgCl_2$, 10 mM HEPES, 0.5 mM EGTA, 2 mM ATP and pH 7.4 adjusted with KOH for the current-clamp mode recording. Firing of EYFP-expressing neurons in the DRN were recorded in the current-clamp mode at a temperature of 30 °C. The combination of a MultiClamp 700B amplifier, Digidata 1440A A/D converter and Clampex 10.3 software (Molecular Devices) was used to trigger the 589 nm-laser for photo-stimulation (950 µW) and control membrane voltage and data acquisition. The spontaneous firing and resting membrane potential were recorded, and the average data for 1 min before the light illumination (Pre), during the 1-min light illumination (ON), and after the termination of light illumination for 1 min (Post) were calculated by Clampfit 10.3 software (Molecular Devices) and Igor Pro (WaveMetrics). Recordings were conducted from five cells from three animals.

**Statistics and reproducibility.** Sample sizes were determined based on previous publications[14,28,29,47]. Animals were assigned randomly to control and experimental groups. Although experimenters were not blinded to group allocation for data collection, subsequent offline analysis of behavioral videos was performed blinded to experimental conditions. The experimenter was blinded to experimental conditions for analysis of immunohistochemistry and in situ hybridization.

GraphPad Prism 9.3.1 software (GraphPad Software Inc.) was used for statistical analysis. Parametric analyses such as ANOVA (two-way and on-way, repeated and non-repeated) and unpaired t test (two-sided) were conducted for most of data, but if the dataset did not follow a normal distribution, nonparametric analysis such as Kruskal–Wallis test, Mann–Whitney test, Wilcoxon matched-pairs signed-rank test and Friedman test were conducted (all two-sided). If the dataset did not have equal variance, repeated-measures ANOVA with the Geisser-Greenhouse correction were used. For ANOVA analyses, Tukey's multiple comparison tests were used as post hoc test, but for two-way repeated-measures ANOVA with the Geisser-Greenhouse correction, a *t* test with Bonferroni's correction was conducted post hoc (all two-sided). For nonparametric analysis, Dunn's multiple comparison test was used as a post hoc test (two-sided). For DRN-VTA ChR2 analysis, one animal had missing data and thus a mixed-effects model was used. All statistical details can be found in the figure legends.

Reproducibility of the effect of optogenetic stimulation was confirmed in two experiments (LHb-DRN ChR2 stimulation and DRN-VTA ChR2 stimulation) by which the same manipulation was conducted independently in two different animal facilities. Also, data were collected using biological replicates (multiple brain slices per animal analyzed for immunohistochemistry and in situ hybridization). Each behavioral data was obtained from biologically independent mice, and the number of animals or cells analyzed in each experiment was presented in the figure legends. Histological verification was conducted in all animals included in this study, and expression patterns of representative images presented in this paper were observed in all other replicates.

**Reporting summary.** Further information on research design is available in the Nature Research Reporting Summary linked to this article.

## Data availability
Source data are provided with this paper.

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

## Acknowledgements

We thank Yumi Asano and Maiko Mitsui for help with animal care and histology. This research was supported by JSPS KAKENHI Grant Numbers JP17H04766, JP19H05202, JP21H00183, Japan Science and Technology Agency (JST) Adaptable and Seamless Technology transfer Program through Target-driven R&D (A-STEP) Grant Number JPMJTM20BW and JST FOREST Program Grant Number JPMJFR214A to A.T., and by National Institute of Mental Health grants R01MH114882-01, R01MH104559, and R01MH127820 to S.J.R.

## Author contributions

A.T. and S.J.R. designed the study. Stereotaxic surgeries and behavior experiments were conducted by A.T., R.D.C., M.E.F., H.A., L.L. and K. Mitsui. IHC analysis and microscopy were performed by A.T. Electrophysiological slice recording of LHb-DRN neuron were conducted by R.D.C. and V.P., under the supervision of R.D.B. Tph2-ArchT transgenic mice were provided by K.F.T. and A.Y., and electrophysiological recoding of serotonin neurons were conducted by E.H., T.T., T.S., K.F.T. and A.Y. AAV production and purification, as well as advised on approaches and data analysis, were provided by Y.C. and M.L. in situ hybridization, microscopy and analyzed data were conducted by K. Miya, T.O., and K. K.-M. Results were analyzed and interpreted by A.T., S.O. and S.J.R. The manuscript was written by A.T. and S.J.R. and edited by all authors.

## Competing interests

The authors declare no competing interests.
