## [Peer Review File · Nature Communications]

Lateral habenula glutamatergic neurons projecting to the dorsal raphe nucleus promote aggressive arousal in miceREVIEWER COMMENTS

Reviewer #1 (Remarks to the Author):

The manuscript "Lateral habenula glutamatergic 1 neurons projecting to the dorsal raphe nucleus promote aggressive arousal" by Aki Takahashi et al. shows that glutamatergic projections from the lateral habenula (LHb) to the dorsal raphe nucleus (DRN) are involved in heightened aggression following social instigation. By using retrobeads and c-fos labeling, the authors show that LHb projection neurons to the DRN are glutamatergic and selectively activated by social instigations. Inhibition of the LHb-DRN projections with chemogenetic and optogenetic approaches prevents escalation of aggression by social instigations, while optogenetic activation of these projections increases aggression. The authors further show that LHb projections mainly innervate non-serotonergic DRN neurons. Optogenetic inhibition of 5-HT neurons in the DRN does not affect aggression escalation induced by social instigation. Finally, the authors show that optogenetic activation of the DRN-VTA projection increases aggression. The authors conclude that the LHb glutamatergic projections to the DRN escalate aggressive behavior by activating non-serotonergic DRN neurons projecting to the VTA. This study identifies a DRN input and the DRN cell type that are involved in social instigation induced aggression escalation. These findings are interesting and improve understanding of the circuit mechanism for the regulation of aggressive behavior. However, this study is preliminary with some major concerns that need to be addressed.

Specific comments:

1. The authors cited their previous microdialysis study which showed that glutamate in the DRN is increased during aggressive behavior and social instigation for their claim that the LHb-DRN projections are activated by social instigation. This study did not specifically measure neural activities of the LHb-DRN projections. Although they used c-fos labeling to show that the DRN projection neurons of the LHb are activated specifically by social instigation, c-fos does not report the specific pattern of neural activities and has a poor temporal resolution. As the LHb-DRN projection is the main topic of this study, it is critical to confirm activation of the LHb-DRN projection neurons, the temporal relationship between LHb neuron activation, social instigation, and aggressive behavior, firing patterns, and the consequence of LHb neuron activation on DRN neuron activities with calcium imaging or electrophysiology.
2. The experiment of optogenetically activating the DRN-VTA projections does not support that the LHb increases aggression through the DRN-VTA projections as there is no evidence that this manipulation mimics the activation of LHb-DRN projections by social instigation. The conclusion based on this experiment that the LHb inputs to the DRN promote aggression by activating VTA projecting DRN neurons is not grounded.
3. The rationale of studying LHb is unclear as DRN receives glutamatergic projections from many brain areas besides LHb, including the prefrontal cortex, hypothalamic areas, and the extended amygdala.
4. Attack latency is only shown in Supplemental Fig 1 and missing from other figures. It is an important component of aggressive behavior that needs to be looked at.
5. The image quality is generally poor. Region of interest (e.g. DRN) should be outlined. High-magnification images should be provided for co-localization experiments in Fig 1f-h, Fig 2C.
6. The authors found that both the ON and ON/ON stimulation schemes increased aggressive behavior compared to OFF sessions, but there is no statistically significant difference between on/off and off (Fig. 4i,j). These results are contradictory to the claim that the LHb-DRN projection is responsible for aggression escalation induced by social instigation, which predicts that the ON/Off scheme should increase aggression.

7. There are some inappropriate statements:

“In particular, continuous activation of the LHb-DRN projection prior to an aggressive encounter caused a strong escalation of aggressive behaviors”. This is not a proper interpretation of Figure 4 as the ON/ON regime does not distinguish if stimulation before or during aggressive encounter elevates aggression.

“Thus, our data show that depending on the subregion of the LHb or neural projection targets, activation of LHb neurons can have different effects on aggressive behavior.” This manuscript does not have data on LHb subregions.

8. In Figure 6G, eArchT-EYFP does not delimit the contour of a cell. How to use YFP signals to determine if mcherry cells are YFP positive?

9. DRN neurons project to many brain regions besides VTA. Why focus on the VTA projections?

10. In the experiment of optogenetic activation of the DRN-VTA projections, an on/off session needs to be added to test if activating these projections mimics social instigation.

11. An off-off-off session should be added to Fig. 7 to test if the decrease over time of the duration of total aggressive behaviors (Fig. 7g) and locomotion (Fig. 7j) is due to habituation effects.

Reviewer #2 (Remarks to the Author):

Takahashi and colleagues investigated the role of lateral habenula (LHb) input to the dorsal raphe (DRN) in mediating aggression. Their results suggest that glutamatergic LHb input to the DRN plays a key role in escalating aggression, but not in basal aggressive behaviour. The authors went on to show that DRN neurons that receive LHb input project to the ventral tegmental area (VTA). Optogenetic activation of the latter projection (DRN-VTA) also increases aggressive behaviour.

Overall, the study is well designed and the topic is timely. The authors have used state-of-the-art techniques and as far as I can see almost all important control experiments were performed to justify the conclusion mentioned above.

I am certain that this study is of interest to a wider community, and thus, the study deserves publication in this journal.

I only have a few minor comments:

- Re Figure 1 – you do give numbers for the Vglut2-positive cells (“we confirmed that 99.2% of RetroBead-positive cells were colocalized with Vglut2 (245 cells among 247 cells analyzed; Fig. 1i-n)...”), but you don’t give any numbers for the cFos-positive neurons in the text. Should be around 20-40 cells according to Figure 1i. Again, assuming these numbers represent all the LHb cells being activated during the task.

- Figure 2f – There is a typo: Frequency. Actually, the typo seems to be present in all figures.

- Re experiments in respect to Figure 4 – here, you stated that you waited 5 weeks for the expression of the constructs. Is there a specific reason for that? I am asking as you mentioned three weeks for the other experiments.

- In terms of the electrophysiological experiments you tested the Arch construct in vitro (Figure 5b), but for the other paradigms you did not test the constructs and basically their efficacy (e.g. for the hM4D-, NpHR- and ChR2-experiments). For example, did the one-minute light stimulation you applied to activate LHb-DRN projections (Figure 4) activate DRN neurons throughout? In other words, does a shorter activation period do the same job?

- Extended Data Figure 3 – does the finding hold true for the right and left LHb? Did you identify any functional asymmetry? Or did you only inject into either the right or left? Please indicate.
- Extended Data Fig. 4e and j – It is difficult to recognise the legend colours as the outline weight seems too large.
- Discussion, line 304-306: "... From these studies, it is clear that the LHb microcircuitry is quite complex and when engaged by aggression can have varying effects depending on the cell type and downstream projection. ..." – I'd suggest to delete 'quite'.

Reviewer #3 (Remarks to the Author):

Takahashi et al. have studied the circuit components behind how social provocation ("instigation") elevates intermale aggression. By combining tracing and opto- and chemogenetic manipulation, they show that LHb-DRN neurons contribute to this escalation effect of prior exposure to (but not direct contact with) another male, but that activity in this pathway does not appear to be related to the baseline levels of aggression (as examined in a resident-intruder paradigm). They identify non-5HT raphe neurons as key to this phenomenon, and further show that a projection from DRN to the VTA is the downstream mediator of instigation-induced escalation of intermale aggression. This is an overall well-designed and -executed study that provides an anatomical framework to the curious – and potentially clinically relevant – role of provocation in augmenting aggression. Several important controls are included (e.g. testing the phenomenon in two separate animal facilities, which I enjoyed seeing presented). There are some issues, esp with regards to with interpretation, that could improve the ms. Some control experiments also need to be added.

MAJOR ISSUES

1. Fig 2, chemogenetic inhibition: I failed to understand if control animals were also injected with CNO (i.e. animals not expressing hM4D)? I.e. is there an effect of CNO independent of the transgene? If not, this is a control that should be included.
2. Fig 4, optogenetic stimulation: the authors hardly discuss the lack of effect in the ON/OFF condition at all. But is this not the crucial experiment? i.e. the activity of the LHb-DRN projection in the period just prior to the introduction intruder is the neuronal correlate to instigation and thus would be expected to yield the same effect as the Inst condition – but it does not. This (negative) result seems as if it argues against the hypothesis. Please address this issue (also in the Results-Discussion) and clarify how it fits into the overall picture.
3. Fig 5b: please include quantification and statistics of the validation of neuronal silencing.
4. Line 123-125: Please clarify how many cells were counted from how many animals. The details on histochemical quantification are overall a bit too brief in this manuscript. This should also be clarified in Fig 4e, f, and in Fig 6g.
5. I am confused by part of the argument the authors are making for the proposed circuit. On p. 12, lines 326-7 it is described that that the DRN target neurons of the LHb projection are serotonergic or GABAergic (I am assuming that these are two separate populations?). But as I understand the model, the LHb and DRN neurons in the LHb-DRN-VTA chain should be excitatory, e.g. to explain why optogenetic stimulation has the effect shown in Fig 7. Can the authors please address this apparent paradox? What is the relationship between the GABAergic and the Vglut3 neurons in the DRN (referenced on line 333-4)?

MINOR ISSUES

6. The manuscript should emphasize for the reader that the authors are examining *intermale* aggression. I understand that for reasons of brevity this is generally shortened to just aggression, but the non-expert reader should be made aware that this is one of several forms of aggression observed in the mouse.

7. Line 55 (and in Discussion): it is not clear to me what justifies the implication of aggressive “arousal” in this context. Please elaborate.
8. p. 9, lines 224-234: why was ArchT used in these experiments and not NpHR3.0 8as in Fig 2)? Please explain rationale.
9. p. 12, lines 323-5 (“The mammalian LHb...”): the message/conclusion of this sentence was not entirely clear to me.
10. p. 14, lines 355-7 (“Thus, it is possible...”): The logic of how instigation should be equivalent to omission of an expected reward is not clear to me. Please elaborate on this argument.
11. p. 14, p. 361-362: (“Although a large body of evidence...”) please add reference(s) for this sentence.
12. p. 14, pg starting on line 372: can the authors please explain how they envision the relationship between instigation and provocation in humans? Are these synonymous terms or is there a difference? If so, what is the difference?
13. Methods: the sources of reagents, equipment etc. is given in an inconsistent manner, sometimes with sometimes without geography, sometimes states are written out in full sometimes in acronyms etc. Please standardize.
14. p. 17, lines 436-8: why were the DRN-VTA ChR2 experiments conducted during the light cycle unlike other experiments? Please explain rationale in the Methods section.
15. p. 18, line 7473: please give concentration and vehicle for Retrobeads.
16. Through the ms. “Resident-Intruder” is sometimes shortened to “RI” (esp Results and Methods), sometime snot. Please revise for consistency.
17. p. 26, Electrophysiology expts (lines 677-86): were solutions bubbled with 100% O₂, not carbogen gas? Please explain why. Also: why was NaHCO₃ not included in the extracellular solution as is standard?
18. Figure legend 1, lines 905-910: “expression” is not an appropriate term to use for the uptake of Retrobeads into a cell. Also in Ext data Fig 2.
19. Ext. data Fig 1: What do the authors mean by the term “Species-typical” in the title of this figure? It is not clear from the data presented.
20. Some grammatical and spelling issues were noted:
 - Line 196: “their” – I assume this refers to the animals but the sentence is not very well constructed.
 - Line 232: “firings” I don’t think this is a term that one would use. Consider e.g. “action potential discharge” or “firing rate”.
 - Line 279: “...light stimulation, was observed...” Remove comma.
 - Line 285: I understand this is up for discussion nowadays but in this referee’s book data is plural.
 - Line 336: “has”, please change to “have”.
 - Line 357: “rewards” should read “reward”.
 - Lines 428-30: “Cob” spelled with one “b” I think?
 - Line 433: “Institute” should read “Institutes”
 - Line 947: “tesst” should read “tests”
 - Figures 2, 3, 5, 7: “Fregency” should read “Frequency”.
 - Ex data Fig 5: “fiver” should read “fiber”

Point-by-point response to the reviewers' comments

We would like to thank the reviewers for their helpful comments on our manuscript. Below is a detailed point-by-point response to the reviewer concerns. We have denoted our responses to the reviewer comments in red text. Changes have been highlighted in yellow throughout the manuscript.

Reviewer #1 (Remarks to the Author):

The manuscript “Lateral habenula glutamatergic 1 neurons projecting to the dorsal raphe nucleus promote aggressive arousal” by Aki Takahashi et al. shows that glutamatergic projections from the lateral habenula (LHb) to the dorsal raphe nucleus (DRN) are involved in heightened aggression following social instigation. By using retrobeads and c-fos labeling, the authors show that LHb projection neurons to the DRN are glutamatergic and selectively activated by social instigations. Inhibition of the LHb-DRN projections with chemogenetic and optogenetic approaches prevents escalation of aggression by social instigations, while optogenetic activation of these projections increases aggression. The authors further show that LHb projections mainly innervate non-serotonergic DRN neurons. Optogenetic inhibition of 5-HT neurons in the DRN does not affect aggression escalation induced by social instigation. Finally, the authors show that optogenetic activation of the DRN-VTA projection increases aggression. The authors conclude that the LHb glutamatergic projections to the DRN escalate aggressive behavior by activating non-serotonergic DRN neurons projecting to the VTA. This study identifies a DRN input and the DRN cell type that are involved in social instigation induced aggression escalation. These findings are interesting and improve understanding of the circuit mechanism for the regulation of aggressive behavior. However, this study is preliminary with some major concerns that need to be addressed.

We really appreciate the reviewer's positive and thoughtful comments that have significantly improved this manuscript.

Specific comments:

1. The authors cited their previous microdialysis study which showed that glutamate in the DRN is increased during aggressive behavior and social instigation for their claim that the LHb-DRN projections are activated by social instigation. This study did not specifically measure neural activities of the LHb-DRN projections. Although they used c-fos labeling to show that the DRN projection neurons of the LHb are activated specifically by social instigation, c-fos does not report the specific pattern of neural activities and has a poor temporal resolution. As the LHb-DRN

projection is the main topic of this study, it is critical to confirm activation of the LHb-DRN projection neurons, the temporal relationship between LHb neuron activation, social instigation, and aggressive behavior, firing patterns, and the consequence of LHb neuron activation on DRN neuron activities with calcium imaging or electrophysiology.

We agree with the reviewer that c-Fos has low temporal resolution and it may not accurately reflect changes in neural firing patterns. As suggested by the reviewer, we initially attempted to record the activity of the LHb-DRN projection neurons using GCaMP, but due to technical difficulties we were unable to record accurate, reliable Ca²⁺ signals within this pathway. As an alternative, we conducted *ex vivo* slice electrophysiology in LHb-DRN projecting neurons and found an increase in firing rate and an increase in resting membrane potential after social instigation-heightened aggression relative to mice that only experienced standard aggression in the resident intruder (RI) test. Our data also showed that social instigation-heightened aggression increased the number of LHb-DRN neurons exhibiting spontaneous activity. We believe that this new data complements and supports existing c-Fos expression data confirming that social instigation-heightened aggression activates the LHb-DRN neurons compared to standard aggression in the RI test. We have now added this data as a new Fig. 2.

2. The experiment of optogenetically activating the DRN-VTA projections does not support that the LHb increases aggression through the DRN-VTA projections as there is no evidence that this manipulation mimics the activation of LHb-DRN projections by social instigation. The conclusion based on this experiment that the LHb inputs to the DRN promote aggression by activating VTA projecting DRN neurons is not grounded.

We thank the reviewer for highlighting this important point. We performed an additional experiment to examine whether activation of the DRN-VTA projection mimics the effect of social instigation. To do this, we stimulated the DRN-VTA projection just prior to RI test to see whether this increases aggressive behavior. We found that ON/OFF stimulation of DRN-VTA projection, as well as ON and ON/ON stimulations, increased intermale aggressive behavior compared to the OFF session. Thus, activation of this pathway mimics the effect of social instigation, suggesting a possible involvement of DRN-VTA projections on instigation-heightened aggression. We now show this data in a new Fig. 8. On the other hand, this is still indirect evidence to conclude that di-synaptic connections of LHb-DRN-VTA are involved in instigation-heightened aggression. Therefore, we mention this important limitation and temper our conclusions by suggesting that the VTA-projecting non-serotonergic DRN neurons may be one of the potential targets, but further work is needed to confirm the di-synaptic nature of this projection (see abstract, introduction, and discussion).

3. The rationale of studying LHb is unclear as DRN receives glutamatergic projections from many brain areas besides LHb, including the prefrontal cortex, hypothalamic areas, and the extended amygdala.

We did analyze c-Fos in several other upstream regions including the lateral hypothalamic area (LH)-DRN projection and did not find evidence that instigation-heightened aggression-activated these relative to a group receiving the standard RI test alone (see Extended Data Fig S2). While we did not analyze prefrontal cortex inputs to DRN, previous studies have shown that the mPFC sends a projection to DRN that promotes social avoidance behavior in the social defeat stress in the mouse (Challis et al 2014 *Front Behav Neurosci*). In the dominant hamster, the mPFC-DRN projection is activated by acute social defeat stress and this promotes a stress resilient phenotype (Gizzell et al 2020 *Front Neural Circuits*). Therefore, neural inputs from the mPFC to the DRN play an important role in social behavior, and we agree that this projection will need to be investigated in future studies to see if they also regulate aggressive behavior. We've included this in the discussion section on page 15.

4. Attack latency is only shown in Supplemental Fig 1 and missing from other figures. It is an important component of aggressive behavior that needs to be looked at.

We thank the reviewer for this suggestion. Attack latency is an important indice of aggressive behavior. We've now added new supplemental Figs (Extended Data Fig. 3, 4, 5, 7, 8, 10, 11) to show the results of all behaviors analyzed including the attack latency. Please note that for some studies we conducted longer habituation sessions before starting optogenetics/chemogenetics experiments in order for animals to show stable aggressive behavior. Many of the animals in these studies with extended habituation showed very short attack latency even without social instigation. Therefore, we could not detect a significant effect of manipulations in this index in some of the experiments.

5. The image quality is generally poor. Region of interest (e.g. DRN) should be outlined. High-magnification images should be provided for co-localization experiments in Fig 1f-h, Fig 2C.

We have increased the image quality, and outlined the region of interest (DRN, LHb). Also, Fig1f-h were substituted with magnified images and an enlarged picture was inserted in Fig. 3c (corresponds to the previous Fig 2c).

6. The authors found that both the ON and ON/ON stimulation schemes increased aggressive behavior compared to OFF sessions, but there is no statistically significant difference between on/off and off (Fig. 4i,j). These results are contradictory to the claim that the LHb-DRN projection

is responsible for aggression escalation induced by social instigation, which predicts that the ON/Off scheme should increase aggression.

This is an important point also raised by Reviewer 3. We have now added an additional experiment to examine the effect of LHb-DRN ON/OFF stimulation in a new batch of animals. We hypothesized that LHb-DRN activation would need to be combined with a social stimulus in order to elicit “aggressive arousal”. Therefore, we conducted a short-term (1 min) social instigation test that we termed subthreshold social instigation, which alone was not enough to produce pro-aggressive effects. Our results show that a combination of subthreshold social instigation paired with optogenetic ON/OFF stimulation of LHb-DRN projection caused a significant increase in aggressive behavior compared to RI test alone or short-term instigation without optogenetic stimulation. We’ve added these new data to Fig. 5o-q, as well as to Extended Data Fig. 7.

7. There are some inappropriate statements:

“In particular, continuous activation of the LHb-DRN projection prior to an aggressive encounter caused a strong escalation of aggressive behaviors”. This is not a proper interpretation of Figure 4 as the ON/ON regime does not distinguish if stimulation before or during aggressive encounter elevates aggression.

We now exclude this sentence from the main text.

“Thus, our data show that depending on the subregion of the LHb or neural projection targets, activation of LHb neurons can have different effects on aggressive behavior.” This manuscript does not have data on LHb subregions.

We agree with the reviewer and have modified the discussion accordingly.

8. In Figure 6G, eArchT-EYFP does not delimit the contour of a cell. How to use YFP signals to determine if mcherry cells are YFP positive?

As the reviewer points out, it is hard to identify EYFP+ cells by using eArchT-EYFP. However, it is still possible to observe the cell-body-like round shape surrounding DAPI, which labels the nuclei. Thus, we counted the EYFP+ cells when the edge of mCherry (or DAPI) were surrounded by EYFP. We also conducted an additional experiment by labeling serotonin neurons with Tph2 to assess its co-localization with EYFP (Extended Data Fig. 9a-e). Higher magnification images are now included in Fig. 7.

9. DRN neurons project to many brain regions besides VTA. Why focus on the VTA projections?

We thank the reviewer for highlighting this point. It was not clear in our original manuscript why we chose to focus on the VTA among other projection areas. Several previous studies have shown that VTA DA is implicated in escalated aggression and aggression reward (for review see de Almeida et al 2005 Eur J Pharmacol, Golden et al 2019 J Neurosci). Further it has been shown that optogenetic activation of VTA dopamine neurons increases aggressive behavior (Yu et al 2014 Mol Psychiatry). From these important studies, as well as our own data, which shows a substantial projection from the DRN to VTA, we chose the VTA as a first target to examine in this study. We've added this rationale to the newly revised manuscript on page 12 and also discuss the necessity of examining other projection targets on aggression in future studies.

10. In the experiment of optogenetic activation of the DRN-VTA projections, an on/off session needs to be added to test if activating these projections mimics social instigation.

We appreciate the reviewer's advice here. As we mentioned in response to point # 2 above, we've added a new set of experiments to examine the effects of ON/OFF stimulation of DRN-VTA projections. Our result show that ON/OFF stimulation indeed increases aggressive behavior, mimicking the effect of social instigation. This data strengthens our proposal that the DRN-VTA projection is involved in instigation-heightened aggression. Please see Fig. 8 for this new data.

11. An off-off-off session should be added to Fig. 7 to test if the decrease over time of the duration of total aggressive behaviors (Fig. 7g) and locomotion (Fig. 7j) is due to habituation effects.

As mentioned in the previous critique, we've now added new data in Fig. 7 that shows no indication of a habituation effect in the new data set.

Reviewer #2 (Remarks to the Author):

Takahashi and colleagues investigated the role of lateral habenula (LHb) input to the dorsal raphe (DRN) in mediating aggression. Their results suggest that glutamatergic LHb input to the DRN plays a key role in escalating aggression, but not in basal aggressive behaviour. The authors went on to show that DRN neurons that receive LHb input project to the ventral tegmental area (VTA). Optogenetic activation of the latter projection (DRN-VTA) also increases aggressive behaviour.

Overall, the study is well designed and the topic is timely. The authors have used state-of-the-art techniques and as far as I can see almost all important control experiments were performed to justify the conclusion mentioned above.

I am certain that this study is of interest to a wider community, and thus, the study deserves publication in this journal.

I only have a few minor comments:

We really appreciate the reviewer's positive and thoughtful comments that have significantly improved this manuscript.

- Re Figure 1 – you do give numbers for the Vglut2-positive cells (“we confirmed that 99.2% of RetroBead-positive cells were colocalized with Vglut2 (245 cells among 247 cells analyzed; Fig. 1l-n)...”), but you don't give any numbers for the cFos-positive neurons in the text. Should be around 20-40 cells according to Figure 1i. Again, assuming these numbers represent all the LHb cells being activated during the task.

We apologize that our analysis was not clear in this Figure. The original data represented in Fig 1 was the average number of c-Fos+ or Retrobead+ cells in one side of the LHb per slice (among 6 consecutive slices with a 90 um interval). We've now re-calculated all analyses to reflect the total number of c-Fos+ or Retrobead+ cells per slice (adding both left and right hemispheres). We have also added new discussion of the number of c-Fos+ cells and Retrobeads+ cells, as well as the % of co-localization in each group in the main text.

- Figure 2f – There is a typo: Frequency. Actually, the typo seems to be present in all figures.

We really appreciate the reviewer catching this typo. It has been corrected in all figures.

- Re experiments in respect to Figure 4 – here, you stated that you waited 5 weeks for the expression of the constructs. Is there a specific reason for that? I am asking as you mentioned three weeks for the other experiments.

For optogenetic experiments with terminal stimulation (LHb-DRN eNpHR3.0 (new Fig. 4), LHb-DRN Chr2 (new Fig. 5), and DRN-VTA Chr2 (new Fig. 8)) we waited 5 weeks post-injection based on our previous studies (Golden et al 2016 Nature; Christoffel et al., Nat Neurosci, 2015). A 4-6 week incubation period is necessary for opsins to be expressed at high enough levels in the terminal for circuit specific stimulation protocols. We have added the following sentence in the method section to explain this: “Behavior tests with optical stimulation was started at least five weeks after AAV injection for optimal expression of the opsin in the projection terminal³³”. By contrast, for DREADD experiments we typically wait 3-4 weeks because we observe strong expression of hM4D-EYFP within the soma at this time point. In fact, as we describe in the methods, the first CNO injection for these studies was administered 4 weeks after the AAV

injection.

- In terms of the electrophysiological experiments you tested the Arch construct in vitro (Figure 5b), but for the other paradigms you did not test the constructs and basically their efficacy (e.g. for the hM4D-, NpHR- and ChR2-experiments). For example, did the one-minute light stimulation you applied to activate LHB-DRN projections (Figure 4) activate DRN neurons throughout? In other words, does a shorter activation period do the same job?

For all other AAV constructs (i.e. hM4D, eNpHR3.0, and ChR2) we used commonly validated parameters for LHB-DRN and DRN-VTA circuits based on many published studies (Atasoy et al 2012 Nature, Anikeeva et al 2011 Nat Neurosci, Okada et al 2014 Nat Commun, Mohan Iyer et al 2014 Nat Biotechnol). We've now added these references in the method section. By contrast, the Tph2-ArchT transgenic mouse line is relatively novel and had not yet been validated for these purposes.

In terms of the length of light stimulation, our data suggests that one-minute light stimulation can activate DRN neurons throughout. As shown in Figure 5o-q, when 1 min stimulation of LHB-DRN was combined with short-term social instigation ("ON/OFF" stimulation scheme), it increased aggressive behavior of male mice similarly to 5 min stimulation ("ON") or 6 min stimulation ("ON/ON") shown in Fig5i.

- Extended Data Figure 3 – does the finding hold true for the right and left LHB? Did you identify any functional asymmetry? Or did you only inject into either the right or left? Please indicate.

We did not observe any functional asymmetry in left and right LHB and thus we combined right and left LHB stimulation in this Figure. We now mention this within the Figure legend (new Extended Data Fig. 6). Also, circuit specific c-Fos assessment showed no difference in c-Fos expression or c-Fos colocalization with RetroBead between right and left LHB. We have now added a sentence in the main text to mention that there was no functional asymmetry in the RetroBeads experiment (Line 139-141).

- Extended Data Fig. 4e and j – It is difficult to recognise the legend colours as the outline weight seems too large.

We've modified the outline weight to have better visibility.

- Discussion, line 304-306: "... From these studies, it is clear that the LHB microcircuitry is quite complex and when engaged by aggression can have varying effects depending on the cell type

and downstream projection. ...” – I’d suggest to delete ‘quite’.

We deleted “quite” from the sentence.

Reviewer #3 (Remarks to the Author):

Takahashi et al. have studied the circuit components behind how social provocation (“instigation”) elevates intermale aggression. By combining tracing and opto- and chemogenetic manipulation, they show that LHb-DRN neurons contribute to this escalation effect of prior exposure to (but not direct contact with) another male, but that activity in this pathway does not appear to be related to the baseline levels of aggression (as examined in a resident-intruder paradigm). They identify non-5HT raphe neurons as key to this phenomenon, and further show that a projection from DRN to the VTA is the downstream mediator of instigation-induced escalation of intermale aggression. This is an overall well-designed and -executed study that provides an anatomical framework to the curious – and potentially clinically relevant – role of provocation in augmenting aggression. Several important controls are included (e.g. testing the phenomenon in two separate animal facilities, which I enjoyed seeing presented). There are some issues, esp with regards to with interpretation, that could improve the ms. Some control experiments also need to be added.

We really appreciate the reviewer’s positive and thoughtful comments that have significantly improved this manuscript.

MAJOR ISSUES

1. Fig 2, chemogenetic inhibition: I failed to understand if control animals were also injected with CNO (i.e. animals not expressing hM4D)? I.e. is there an effect of CNO independent of the transgene? If not, this is a control that should be included.

The control EYFP animals also received CNO at the same time as hM4D-expressing animals. Because CNO injection (3rd session (Inst) or 7th session (RI)) did not elicit any significant differences in behavior compared to saline injection (2nd & 5th sessions (Inst) or 6th session (RI)) in the control animal, we concluded that there was no effect of CNO on aggressive behaviors.

2. Fig 4, optogenetic stimulation: the authors hardly discuss the lack of effect in the ON/OFF condition at all. But is this not the crucial experiment? i.e. the activity of the LHb-DRN projection in the period just prior to the introduction intruder is the neuronal correlate to instigation and thus would be expected to yield the same effect as the Inst condition – but it does not. This (negative) result seems as if it argues against the hypothesis. Please address this issue (also in

the Results-Discussion) and clarify how it fits into the overall picture.

We thank the reviewer for pointing this out. This is an important point also raised by Reviewer 1. Please see response #2 to Reviewer 1.

3. Fig 5b: please include quantification and statistics of the validation of neuronal silencing.

We have now conducted a new set of slice recordings to confirm the eArchT inhibition of serotonergic neurons (5 cells from 3 animals). We confirmed that the yellow light illumination significantly reduced spontaneous firing and reduced resting membrane potential. Also, we show that eArchT+ cells recovered spontaneous firing immediately after the 10 min of light illumination (Fig 6a-c). Because the recording protocol was modified for this new experiment (i.e. from loose cell attached recordings to whole cell patch-clamp recording), we've altered the method section accordingly.

4. Line 123-125: Please clarify how many cells were counted from how many animals. The details on histochemical quantification are overall a bit too brief in this manuscript. This should also be clarified in Fig 4e, f, and in Fig 6g.

We've added additional details about the histochemical quantification in the results and methods section, including the number of animals and cells per animal used for the analysis (Line 128-138 for Fig. 1i,j,k; Line 225-231 for Fig. 5e,f (corresponds to previous Fig. 4e,f); Line 306-307 for Fig. 7g (corresponds to previous Fig. 6g) and Line 311-317 for Extended Data Fig. 9).

5. I am confused by part of the argument the authors are making for the proposed circuit. On p. 12, lines 326-7 it is described that that the DRN target neurons of the LHb projection are serotonergic or GABAergic (I am assuming that these are two separate populations?). But as I understand the model, the LHb and DRN neurons in the LHb-DRN-VTA chain should be excitatory, e.g. to explain why optogenetic stimulation has the effect shown in Fig 7. Can the authors please address this apparent paradox? What is the relationship between the GABAergic and the *Vglut3* neurons in the DRN (referenced on line 333-4)?

We apologize for our misleading description. In addition to serotonergic and GABAergic neurons, the DRN contains glutamatergic, dopaminergic, and peptidergic neurons. In this study, we show that a majority of DRN neurons that receive LHb input were not *Tph2+* (Fig. 7a-g, Extended Data Fig. 9a-e) or *TH+* (Extended Data Fig 9f-j). To examine if LHb-DRN-VTA is excitatory, we have include an additional experiment to determine whether DRN neurons receiving LHb input co-localize with *Vglut3* as a marker of glutamatergic neurons in the DRN. Our data shows that

about 44% of DRN neurons that received LHb input are *Vglut3*-positive neurons (new Extended Data Fig 9k-p). This was consistent with previous studies showing that the major populations of VTA-projecting DRN neurons (~46%) were *Vglut3*⁺/*Tph*⁻ (Qi et al 2014 Nat Commun). In addition, we confirmed that DRN-VTA stimulation increases *c-Fos* expression in the VTA (new Fig. 8d). Therefore, these results suggest that LHb-DRN-VTA circuit is excitatory.

MINOR ISSUES

6. The manuscript should emphasize for the reader that the authors are examining *intermale* aggression. I understand that for reasons of brevity this is generally shortened to just aggression, but the non-expert reader should be made aware that this is one of several forms of aggression observed in the mouse.

We now state “intermale aggression” throughout the manuscript. We have also added new data where we examine whether activation of LHb-DRN projection triggers aggression towards females, but found no evidence for female-directed aggression (new Fig 5r).

7. Line 55 (and in Discussion): it is not clear to me what justifies the implication of aggressive “arousal” in this context. Please elaborate.

We thank the reviewer for bringing up this important point. We operationally defined the internal state that increases aggressive behavior by social instigation as “aggressive arousal”, based on the definition originally proposed by Michael Potegal. “Aggressive arousal” has been considered as theoretically distinguishable from “general arousal” (proposed as behavior-specific drive by D.E. Berlyne), as it specifically affects aggressive components of behaviors but not other behaviors reflective of general arousal states, such as locomotor activity or sexual behavior. To examine the effect of activation of the LHb-DRN projection in greater detail, we examined whether stimulation affects locomotor activity or female-directed behavior. Our results show that LHb-DRN stimulation did not increase locomotor activity, nor did it induce sexual behavior towards a female (new Fig. 5r, Extended Data Fig. 7j). Therefore, activation of LHb-DRN projection specifically enhances intermale aggression without affecting other arousal-related behaviors. We provide a more sophisticated explanation of aggressive arousal in the introduction (Line 85-86).

8. p. 9, lines 224-234: why was ArchT used in these experiments and not NpHR3.0 8as in Fig 2)? Please explain rationale.

Previous studies have shown that ArchT is a better opsin for long-term neural silencing than eNpHR3.0 (Chow et al 2010 Nature, Tsunematsu et al 2013 Behav Brain Res). Because this study

aimed to suppress 5-HT neural activity for up to 10 min (5 min Inst + 5 min RI), we decided to use eArchT3.0. Indeed, our electrophysiological recording shows successful inhibition of 5-HT neurons for 10 min by eArchT3.0 illumination (Fig. 6). By contrast, for the optogenetic inhibition of projection terminals, it has shown that ArchT can have a paradoxical excitatory effect (Mahn et al 2016 Nat Neurosci) and NpHR3.0 is considered to be desirable for this purpose. Therefore, we used NpHR3.0 for LHB-DRN terminal inhibition experiments with previously published illumination protocols (Fig. 4). This is also now described in the method section.

9. p. 12, lines 323-5 (“The mammalian LHB...”): the message/conclusion of this sentence was not entirely clear to me.

We have modified the text to clarify our conclusion.

10. p. 14, lines 355-7 (“Thus, it is possible...”): The logic of how instigation should be equivalent to omission of an expected reward is not clear to me. Please elaborate on this argument.

Previous studies have shown that the expression of aggressive behavior is rewarding to male mice and in the social instigation test, the resident male’s motivation to express aggressive behavior towards an intruding rival is hampered by the existence of the protective cage, which might be similar to an omission of the reward. It has been shown that omission of expected rewards (either food or water) causes escalation of aggressive behavior in the mouse and pigeon. Although this is just speculation, we think this possibility will be interesting to examine in future studies. We have now included this elaborated discussion, however, we are also happy to remove it if the reviewer does not feel it is warranted.

11. p. 14, p. 361-362: (“Although a large body of evidence...”) please add reference(s) for this sentence.

This has been added.

12. p. 14, pg starting on line 372: can the authors please explain how they envision the relationship between instigation and provocation in humans? Are these synonymous terms or is there a difference? If so, what is the difference?

While instigation and provocation are thought to tap into similar processes, there are some key differences that we now highlight in the manuscript. Commonly used methods for provocation in human studies within a laboratory setting involve the test subject receiving punishment (i.e. electrical shock or point subtraction) from a fictitious opponent in a competitive game. In the

social instigation procedure, animals encounter an opponent via sensory contact but the test subject does not receive punishment *per se*. Both provocation and instigation involve a potentially hostile rival, and both increase aggressive behavior. We have modified the discussion to describe this potential relationship.

13. Methods: the sources of reagents, equipment etc. is given in an inconsistent manner, sometimes with sometimes without geography, sometimes states are written out in full sometimes in acronyms etc. Please standardize.

This has been corrected for consistency

14. p. 17, lines 436-8: why were the DRN-VTA ChR2 experiments conducted during the light cycle unlike other experiments? Please explain rationale in the Methods section.

As the reviewer pointed out, this experiment was conducted in the ISMMS facility where the light-dark cycle was not reversed. Because one reviewer asked to examine the effect of ON/OFF stimulation of DRN-VTA projection to determine whether this projection is capable of mimicking the effect of social instigation, we conducted a new experiment to examine the effect of ON, ON/OFF, and ON/ON stimulation schemes of DRN-VTA projection in the dark cycle in the animal facility of the University of Tsukuba. We found that activation of the DRN-VTA projection also increased intermale aggressive behavior in the dark cycle confirming our earlier studies that were performed during the light phase. We have added this data to Fig 8.

15. p. 18, line 7473: please give concentration and vehicle for Retrobeads.

We did not dilute Retrobeads in this study.

16. Through the ms. "Resident-Intruder" is sometimes shortened to "RI" (esp Results and Methods), sometime snot. Please revise for consistency.

We have now corrected this throughout the manuscript for consistency

17. p. 26, Electrophysiology expts (lines 677-86): were solutions bubbled with 100% O₂, not carbogen gas? Please explain why. Also: why was NaHCO₃ not included in the extracellular solution as is standard?

We have added a new experiment using slice electrophysiology to validate appropriate silencing within ArchT-positive DRN neurons (new Fig 6a-c). We also modified the methods to reflect that

the recording conditions were performed in a standard NaHCO₃-containing buffer that was bubbled with 95% O₂ and 5% CO₂.

18. Figure legend 1, lines 905-910: “expression” is not an appropriate term to use for the uptake of Retrobeads into a cell. Also in Ext data Fig 2.

We agree with the reviewer and now changed the word “expression” to “labeling”.

19. Ext. data Fig 1: What do the authors mean by the term “Species-typical” in the title of this figure? It is not clear from the data presented.

We used the term “species-typical aggression” to indicate resident-intruder aggressive behavior because we operationally defined the resident-intruder aggressive behavior without social instigation as the species-typical level of aggressive behavior in this study. However, to be more accurate, we now changed it to “resident-intruder aggression”.

20. Some grammatical and spelling issues were noted:

Line 196: “their” – I assume this refers to the animals but the sentence is not very well constructed.

Line 232: “firings” I don’t think this is a term that one would use. Consider e.g. “action potential discharge” or “firing rate”.

Line 279: “...light stimulation, was observed...” Remove comma.

Line 285: I understand this is up for discussion nowadays but in this referee’s book data is plural.

Line 336: “has”, please change to “have”.

Line 357: “rewards” should read “reward”.

Lines 428-30: “Cob” spelled with one “b” I think?

Line 433: “Institute” should read “Institutes”

Line 947: “tesst” should read “tests”

Figures 2, 3, 5, 7: “Frequency” should read “Frequency”.

Ex data Fig 5: “fiver” should read “fiber”

We appreciate the reviewer’s careful reading of our manuscript. We have fixed all the typos and errors.

REVIEWER COMMENTS

Reviewer #1 (Remarks to the Author):

The authors have adequately addressed my comments. A few minor issues raised by Reviewer 3 should be further addressed before publication.

Reviewer 3's comment 4: Line 123-125: Please clarify how many cells were counted from how many animals. The details on histochemical quantification are overall a bit too brief in this manuscript. This should also be clarified in Fig 4e, f, and in Fig 6g.

Line 131-133 now reads as "Both right and left hemispheres of
132 the LHb from 6 consecutive slices with a 90 μ m interval were analyzed in control (Cont, n = 8), RI (n = 8), and Inst (n = 10) groups.

It should give the number of animals from which to get the 6 slices. If control, n=8 means 8 animals, why there are only 6 slices. If it is 6 slices from 8 animals, it should be written as 48 slices from 8 animals.

Description of the new results in line 136-146 should have slice and animal numbers.

Reviewer 3' comment 10. p. 14, lines 355-7 ("Thus, it is possible..."): The logic of how instigation should be equivalent to omission of an expected reward is not clear to me. Please elaborate on this argument.

The discussion to connect instigation and omission of expected rewards is tenuous. It should be removed.

Reviewer 3's comment 12. p. 14, pg starting on line 372: can the authors please explain how they envision the relationship between instigation and provocation in humans? Are these synonymous terms or is there a difference? If so, what is the difference?

Human aggression induced by provocation and social instigation-induced aggression in this study have different theoretical models. It is not appropriate to assume they have same brain mechanisms. The paragraph on human provocation should be removed or replaced with a human condition that is truly relevant to social instigation.

Reviewer #2 (Remarks to the Author):

I am satisfied with the changes made by the authors.

Reviewer #3 (Remarks to the Author):

The authors have carefully attended to the concerns I raised, and introduced several new experiments and analyses. The manuscript is, as a result, improved and will be a very valuable addition to the literature that I think will attract the interest of a large audience. The authors are congratulated on this fine work.

Point-by-point response to the reviewers' comments

Reviewer #1 (Remarks to the Author):

The authors have adequately addressed my comments. A few minor issues raised by Reviewer 3 should be further addressed before publication.

We really appreciate the reviewers' thoughtful comments that have helped to improve readability of our manuscript. Importantly, we have removed discussion about the relevance of our findings to aggression instigation in humans.

Reviewer 3's comment 4: Line 123-125: Please clarify how many cells were counted from how many animals. The details on histochemical quantification are overall a bit too brief in this manuscript. This should also be clarified in Fig 4e, f, and in Fig 6g.

Line 131-133 now reads as "Both right and left hemispheres of 132 the LHb from 6 consecutive slices with a 90 μ m interval were analyzed in control (Cont, n = 8), RI (n = 8), and Inst (n = 10) groups.

It should give the number of animals from which to get the 6 slices. If control, n=8 means 8 animals, why there are only 6 slices. If it is 6 slices from 8 animals, it should be written as 48 slices from 8 animals.

Description of the new results in line 136-146 should have slice and animal numbers.

We apologize for the confusion. We've now modified description of our methods as follows: "Both right and left hemispheres of the LHb from 6 consecutive slices with a 90 μ m interval were analyzed from each animal in control (Cont, n = 8 animals), RI (n = 8 animals), and Inst (n = 10 animals) groups. The average number of c-Fos- or Retrobead-labeled cells and their % colocalization in the LHb per slice were calculated in each animal."

Reviewer 3' comment 10. p. 14, lines 355-7 ("Thus, it is possible..."): The logic of how instigation should be equivalent to omission of an expected reward is not clear to me. Please elaborate on this argument.

The discussion to connect instigation and omission of expected rewards is tenuous. It should

be removed.

We have now removed the entire paragraph describing connections between instigation and omission of expected rewards.

Reviewer 3's comment 12. p. 14, pg starting on line 372: can the authors please explain how they envision the relationship between instigation and provocation in humans? Are these synonymous terms or is there a difference? If so, what is the difference?

Human aggression induced by provocation and social instigation-induced aggression in this study have different theoretical models. It is not appropriate to assume they have same brain mechanisms. The paragraph on human provocation should be removed or replaced with a human condition that is truly relevant to social instigation.

We agree and have removed this paragraph.

Reviewer #2 (Remarks to the Author):

I am satisfied with the changes made by the authors.

We really appreciate the reviewer's positive and thoughtful comments throughout the review process.

Reviewer #3 (Remarks to the Author):

The authors have carefully attended to the concerns I raised, and introduced several new experiments and analyses. The manuscript is, as a result, improved and will be a very valuable addition to the literature that I think will attract the interest of a large audience. The authors are congratulated on this fine work.

We really appreciate the reviewer's positive and thoughtful comments throughout the review process.